# Differentiating *Drosophila* female germ cells initiate Polycomb silencing by regulating PRC2-interacting proteins

**Steven Z DeLuca, Megha Ghildiyal[†], Liang-Yu Pang, Allan C Spradling***

Howard Hughes Medical Institute Research Laboratories Department of Embryology, Carnegie Institution for Science, Baltimore, United States

**Abstract** Polycomb silencing represses gene expression and provides a molecular memory of chromatin state that is essential for animal development. We show that *Drosophila* female germline stem cells (GSCs) provide a powerful system for studying Polycomb silencing. GSCs have a non-canonical distribution of PRC2 activity and lack silenced chromatin like embryonic progenitors. As GSC daughters differentiate into nurse cells and oocytes, nurse cells, like embryonic somatic cells, silence genes in traditional Polycomb domains and in generally inactive chromatin. Developmentally controlled expression of two Polycomb repressive complex 2 (PRC2)-interacting proteins, Pcl and Scm, initiate silencing during differentiation. In GSCs, abundant Pcl inhibits PRC2-dependent silencing globally, while in nurse cells Pcl declines and newly induced Scm concentrates PRC2 activity on traditional Polycomb domains. Our results suggest that PRC2-dependent silencing is developmentally regulated by accessory proteins that either increase the concentration of PRC2 at target sites or inhibit the rate that PRC2 samples chromatin.

**\*For correspondence:**
spradling@carnegiescience.edu

**Present address:** [†]Illuminia Corp., Research Division, San Diego, United States

**Competing interests:** The authors declare that no competing interests exist.

## Introduction

Differentiation is the defining mechanism enabling the evolution and development of multicellular animals. During early *Drosophila* embryonic development, cascades of transcription factors transform two initial body axes into a precise coordinate system that identifies nearly every cell by a unique combination of factors based on their position (*Fowlkes et al., 2008*; *Karaiskos et al., 2017*; *St Johnston and Nüsslein-Volhard, 1992*). Further elaboration of a differentiation program requires the acquisition of a cellular 'memory' mediated by an exceptional form of repression known as Polycomb silencing (*Jones and Gelbart, 1990*; *Struhl and Akam, 1985*; *Wedeen et al., 1986*). Initially characterized by genetic studies of Hox gene regulation along the anterior-posterior axis of the *Drosophila* embryo (*Lewis, 1978*), Polycomb group gene (PcG-gene) products recognize repressed loci, coat kilobases of repressed enhancer regions (PcG domains), limit transcription, and restrict eventual cell fates (*Schuettengruber et al., 2017*). Subsequent research revealed that Polycomb silencing is also utilized by mammalian embryos and likely by all animals, and contributes to the differentiation of all somatic embryonic cells as well as progeny cells downstream from pluripotent embryonic stem cells (ESCs) (*Aloia et al., 2013*; *Montgomery et al., 2005*).

The development of germ cells also involves highly regulated changes in gene expression and chromatin organization that differ in important ways from other embryonic cells. Female germ cells in *Drosophila*, mouse, and diverse other species (*Lei and Spradling, 2016*; *Matova and Cooley, 2001*), not only give rise to oocytes but also mostly produce a late-differentiating cell type known as nurse cells that nourish the oocytes by donating cytoplasmic organelles, RNAs, and proteins before undergoing programmed cell death. In *Drosophila*, germ cells undergo four rounds of synchronous divisions to produce interconnected 16-cell germline cysts, which are wrapped by somatic cells to constitute an ovarian follicle. The initial cyst cell becomes the oocyte, whereas the other 15

differentiate as nurse cells (*de Cuevas and Spradling, 1998*). Whether nurse cells are true somatic cells that differentiate using canonical Polycomb silencing or whether they remain germ cells is controversial (*Figure 1A*), because germline mutation of only a few PcG-group genes (*E(z)* and *Su(z)12*) disrupt nurse cell/oocyte differentiation while mutation of most others do not (*Iovino et al., 2013*).

PcG genes encode two major protein complexes, Polycomb Repressive Complex 1 (PRC1), an E3 ligase that ubiquitylates histone H2A on lysine 119 (H2AK119ub) (*Franke et al., 1992*; *Shao et al., 1999*; *Wang et al., 2004*), and PRC2, which methylates histone H3 on lysine 27 (H3K27me1/2/3) (*Cao et al., 2002*; *Czermin et al., 2002*; *Kuzmichev et al., 2002*; *Müller et al., 2002*). In fly embryos, silencing and PRC2-dependent H3K27me3 enrichment on PcG domains first appears as anterior-posterior patterning is being completed during a massive upregulation of zygotic gene expression during cell cycle 14 (*Alhaj Abed et al., 2018*; *Li et al., 2014*; *Pelegri and Lehmann, 1994*). In preimplantation mouse embryos, H3K27me3 is distributed in a 'noncanonical' pattern throughout gene deserts and inactive loci (*Liu et al., 2016*; *Zheng et al., 2016*) and is not further enriched on its canonical sites: CpG-rich 'islands' (CGIs) around enhancers and promoters of developmentally-induced genes (*Ku et al., 2008*; *Tanay et al., 2007*). However, H3K27me3 is highly enriched on CGIs in mouse embryonic stem cell cultures derived from preimplantation embryos (*Boyer et al., 2006*). The transition from the noncanonical to canonical H3K27me3 pattern likely involves the initial PRC2 specificity-establishing events, but this transition remains poorly understood. Unfortunately, studying the establishment of PRC2 specificity in early embryos is hampered by difficulties in purifying and genetically manipulating a short-lived transition state controlled by both a maternal and zygotic supply of PcG proteins. Nevertheless, many studies have investigated factors controlling PcG targeting specificity in other cell types in both flies and mammals, which offer clues into how PcG is established in early embryos (recently reviewed in *Kassis et al., 2017*; *Kuroda et al., 2020*; *Laugesen et al., 2019*; *Yu et al., 2019*). In flies, Polycomb Response Elements (PREs) embedded within PcG domains recruit the DNA-binding PhoRC complex, PRC1, and PRC2, as well as Scm, which physically interacts with all three complexes (*Brown et al., 1998*; *Busturia et al., 2001*; *Fritsch et al., 1999*; *Kang et al., 2015*; *Klymenko et al., 2006*; *Mishra et al., 2001*; *Shimell et al., 2000*). PREs are enriched in several hundred broad H3K27me3-coated loci including the Hox clusters and the enhancers of other developmentally regulated transcription factors and signaling components (*Schwartz et al., 2006*). Mammals, however, lack PRE-like sequences and instead enrich PcG proteins on CGIs to silence a larger number of tissue-specific genes than flies (*Jermann et al., 2014*; *Ku et al., 2008*; *Lynch et al., 2012*; *Mendenhall et al., 2010*; *Tanay et al., 2007*). Regardless of these differences, many PcG proteins are evolutionarily conserved, including factors that were proposed to direct PcG proteins to their target sites. For example, flies encode Jarid2 and Pcl, homologues of mammalian PRC2 accessory proteins that increase PRC2 activity at CGIs by binding to H2AK119ub, other modified histones, or CGI DNA (*Ballaré et al., 2012*; *Cooper et al., 2016*; *Healy et al., 2019*; *Hunkapiller et al., 2012*; *Højfeldt et al., 2019*; *Kalb et al., 2014*; *Li et al., 2010*; *Li et al., 2017*; *Perino et al., 2018*; *Son et al., 2013*; *Walker et al., 2010*). However, H2AK119ub is minimally important for PRC2 targeting in flies (*Gutiérrez et al., 2012*; *Pengelly et al., 2015*), and while fly Pcl promotes PRC2 activity on PREs, it lacks DNA-binding site specificity (*Choi et al., 2017*; *Nekrasov et al., 2007*; *Savla et al., 2008*). How Jarid2, Pcl, or other PcG proteins contribute to PRC2 enrichment and silencing at specific sites, and how PRC2 activity is developmentally controlled are imperfectly understood. Studying how PcG proteins interact with each other and their targets in PcG domains at the time cellular 'memories' are initially formed is critically important to addressing these questions.

We have established a new system for investigating the initiation of Polycomb silencing during development using reporter genes that measure repression at thousands of locations throughout the genome with single-cell resolution, and applied this system to analyze *Drosophila* female germ cell differentiation. *Drosophila* female germline stem cells lack silencing and contain a non-canonical H3K27me3 pattern similar to early embryos. In contrast, nurse cells, during their differentiation from progenitors, acquire canonical Polycomb silencing on similar sites as embryonic somatic cells. Complete silencing of PcG domains in nurse cells involves multiple PcG proteins, including components of PRC1, in addition to core subunits of PRC2. PcG gene mutations are less disruptive in germ cells compared to embryonic cells, because interfering with the single, relatively simple nurse cell program affects oocyte completion more weakly than disrupting myriad, interdependent somatic cell type differentiation programs affects a developing embryo. Finally, we show how two

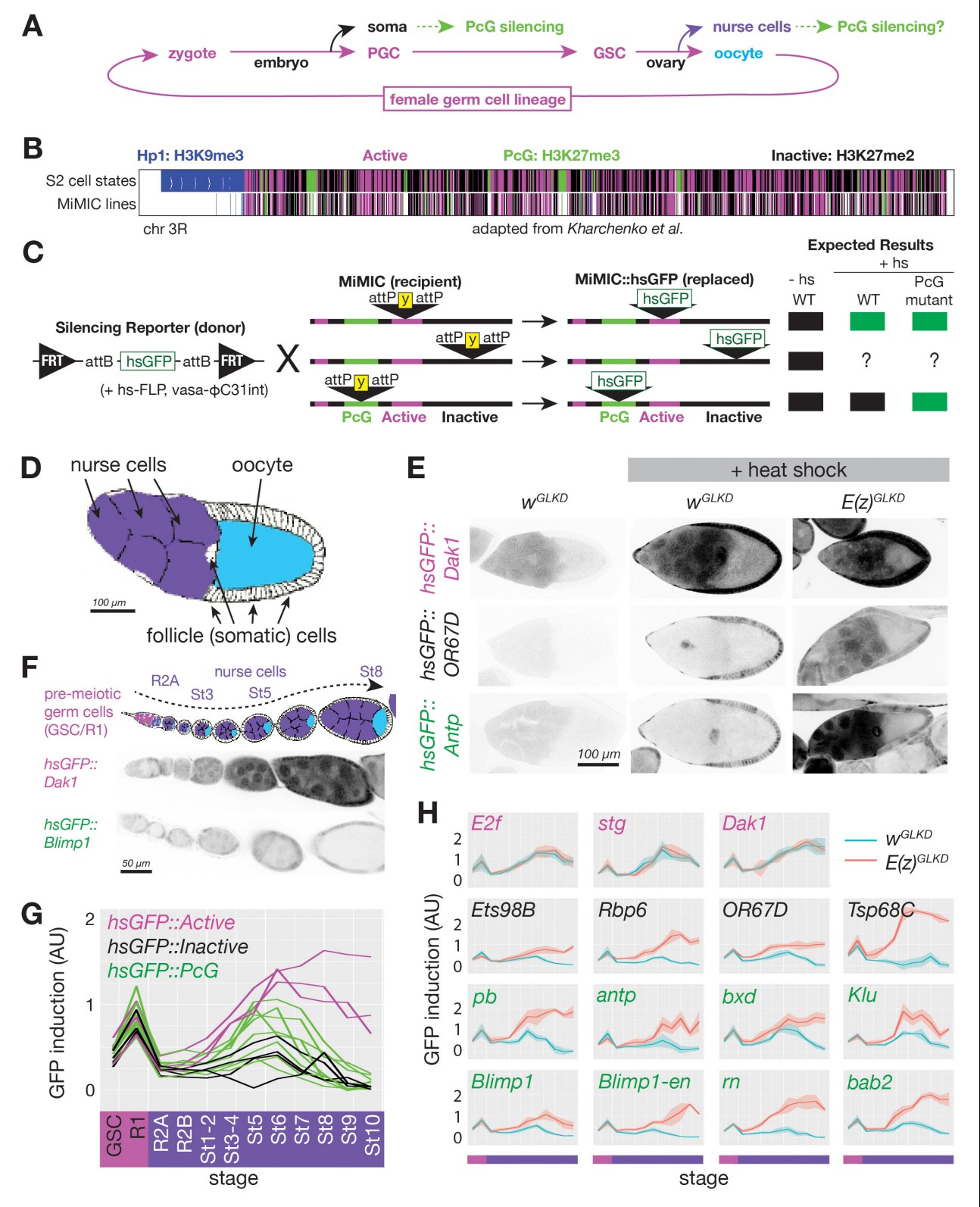

**Figure 1.** Developmentally regulated silencing in inactive and PcG domains in the germline. (**A**) Cyclical lineage of female germ cells and two dead-end derivative lineages, soma and nurse cells. (**B**) Map of chromosome 3R showing color-coded, simplified chromatin states. (**C**) Integration protocol of hsGFP silencing reporters into MiMIC insertions within different chromatin domains and expected expression +/- heat shock (hs). (**D**) Cell types in a stage 10 follicle. Germline knockdown (GLKD) should affect nurse cells (purple), and oocytes (cyan) but not surrounding somatic cells (white). (**E**) Stage

*Figure 1 continued on next page*

*Figure 1 continued*

9/10 follicles showing GFP fluorescence from reporters integrated in active (near *Dak1*), inactive (near *OR67D*) or PcG (near *Antp*) chromatin, in control (*w*$^{GLKD}$), *Scm*$^{GLKD}$, or *E(z)* $^{GLKD}$. Somatic follicle cells serve as an internal control. (F) Diagram of germline development from pre-meiotic stages (GSC/R1, pink). Nurse cells (purple) differentiate from oocytes (cyan) in region 2A (R2A); nurse cells and oocytes grow further (St3-St8). Below, GFP fluorescence after heat shock from two indicated lines. (G) Plot of mean GFP induction ([GFP]$^{+hs}$ – [GFP]$^{-hs}$) in nurse cells or their precursors across 12 developmental stages for 15 reporter lines colored according to their chromatin domain. (H) The effect of *E(z)*$^{GLKD}$ on reporters near the indicated genes colored by domain type. Solid line indicates mean fluorescence; shading shows one standard deviation from the mean. X-axes colored for stage as in G. Size bars: D, E 100 μm; F 50 μm.

The online version of this article includes the following source data for figure 1:

**Source data 1.** Fluorescene intensity measurements for *hsGFP* reporters in *Figure 1G-H* and *Figure 6A*.

developmentally regulated PcG proteins alter PRC2 distribution to initiate silencing during differentiation. Our results suggest a specific model for the establishment of Polycomb silencing in naive precursors, and provide new insights into how PRC2 and related methylases may regulate gene silencing during development. Thus, analyzing the *Drosophila* female germline avoids the cellular and genetic complexity of early embryonic development, and holds great promise for studying many aspects of chromatin regulation.

## Results

### A system of reporters to analyze developmental gene silencing

The unprecedented facility with which the precisely annotated *Drosophila* genome can be manipulated (*Nagarkar-Jaiswal et al., 2015*) encouraged us to develop a method to measure silencing at specific sites throughout the genome in single cells. The idea was to place a single universal reporter gene in many regions of interest and then at each site record how the local chromatin environment changes with time in cells of interest by measuring its effects on the reporter gene. Reporters have proved useful in the past for studying PcG-silenced and Hp1-silenced chromatin in vivo (*Babenko et al., 2010*; *Fitzgerald and Bender, 2001*; *Wallrath and Elgin, 1995*; *Yan et al., 2002*). However, existing reporters were not suitable for probing repressive domains in germ cells for a number of technical reasons. Therefore, we developed a new reporter compatible with female germ cells and an efficient, general method for targeting it to potentially silenced loci. Our reporter (hsGFP) consists of a minimal fragment of the Hsp70A gene containing a heat-shock-inducible enhancer, promoter, and short 5'UTR fused to Green Fluorescent Protein (GFP) and a transcriptional terminator (*Figure 1C*). We chose the heat-shock enhancer and promoter because of its low basal activity, robust inducibility in nearly all cells types, and similarity to promoters of developmentally activated genes (*Guertin et al., 2010*; *Muse et al., 2007*; *Zeitlinger et al., 2007*). Importantly, our reporter was insensitive to ovarian small RNAs derived from *Hsp70* loci due to a truncated 5'UTR (*DeLuca and Spradling, 2018*).

We integrated the reporter into pre-selected MiMIC transposons using FLP and phiC31-catalyzed site-specific recombination (*Nagarkar-Jaiswal et al., 2015*; *Figure 1B,C*). We initially targeted 300 of more than 6000 potential MiMIC sites based on their location within three types of potentially silenced chromatin– an H3K27me3-enriched 'PcG' type (*Figure 1B* green), an H3K9me3enriched 'Hp1' type (*Figure 1B* blue), and an H3K27me2-enriched generally 'inactive' type (*Figure 1B* black), as well as a single type of 'active' chromatin depleted for H3K9 or H3K27 methylation (*Figure 1B* magenta) (modified from *Kharchenko et al., 2011*). Based on publicly available MiMICs on chromosomes 2 and 3 (in 2016), we targeted every MiMIC within a PcG domain on chromosome three and within an Hp1 domain on either chromosome. In addition, we targeted a selection of MiMIC insertions in active and inactive chromatin domains evenly spaced along chromosome 3. From the 300 sites targeted, we successfully isolated at least one reporter integration in 109 sites. In this paper, we present quantitative data from all 12 PcG domain-localized lines as well as 3 and 5 representative lines in active and inactive domains, respectively. Data from Hp1 domain-localized lines will be reported elsewhere.

Our reporter enables simple tests for PcG-repressed chromatin at specific reporter insertion sites, with single-cell resolution, and at nearly every stage of development. While the very different sizes

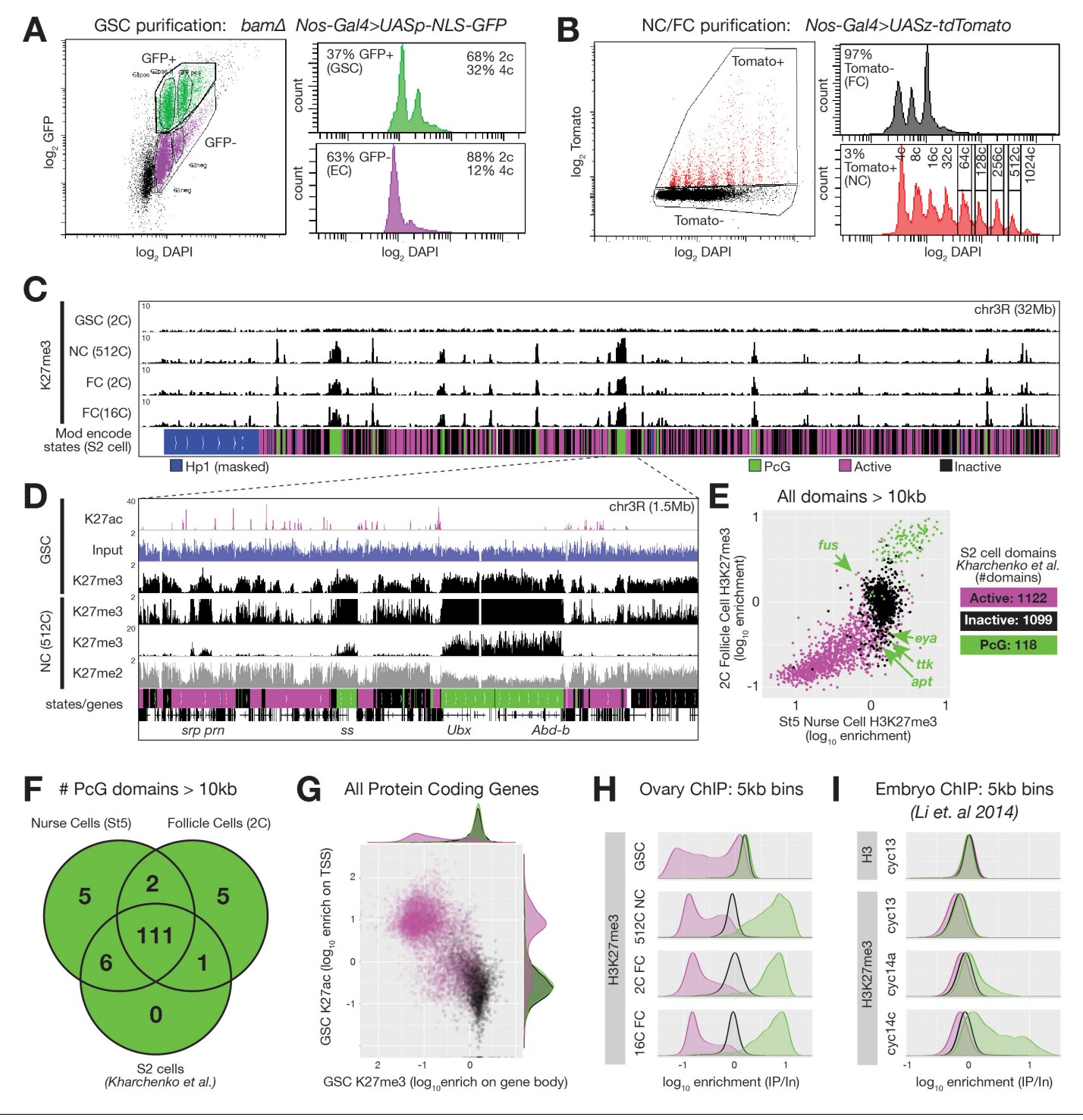

**Figure 2.** ChIPseq of FACS-purified germline chromatin. (A,B) FACSdiva-generated summaries of FACS-sorted fixed nuclei. (A) Nuclei were sorted using GFP from *bam* ovaries expressing germline-specific nuclear GFP vs DNA content (DAPI), yielding GSCs and somatic escort cells (EC). (B) Nurse and Follicle cell nuclei were sorted using Tomato from ovaries expressing germline-specific tdTomato vs. DNA content (DAPI). The haploid DNA content (C-value) is noted above each peak. 2c nuclei were not on scale to aid visualization of larger nurse cells. (C) Chromosome 3R genome browser view of RPM-normalized H3K27me3 ChIPseq read depth from the indicated purified nuclei. Below: chromatin states in S2 cells. (D) Chromosome 3R subregion (dashed lines) showing RPM-normalized read depth from input (blue) or ChIPseq of the indicated epitopes. Nurse cell H3K27me3 read depth is plotted on two scales to show enrichment over a 100-fold range. (E) 2c follicle cell vs. stage 5 nurse cell H3K27me3 enrichment (IP/Input) on every annotated active, inactive, and PcG domain larger than 10 kilobases in S2 cells. PcG domains uniquely depleted of H3K27me3 in nurse cells or follicle

*Figure 2 continued on next page*

**Figure 2 continued**

cells are indicated with an arrow and the name of a gene within the domain. (**F**) Summary of the number of PcG domains shared by follicle cells, nurse cells, and S2 cells. (**G**) Summary of GSC ChIP plotting H3K27ac enrichment (IP/Input) in a 500 bp bin downstream from the annotated transcription start site (TSS) of every protein-coding gene vs. H3K27me3 enrichment (IP/Input) on its gene body. (**H–I**) H3K27me3 enrichment histograms across active (magenta), inactive (black), and PcG (green) domains divided into 5 kb bins tiling the genome. (**H**) Purified ovarian cell types. (**I**) Cycle 13 and 14 embryos (*Li et al., 2014*). In (**I**), total H3 is included as a control to show that active domains are not depleted of H3K27me3 because they are depleted of total H3.

and metabolic activities of different cell types may influence the amount of GFP produced from reporters, our system should reliably detect PcG silencing as long as comparisons are made in a single cell type and developmental stage between reporters in PcG versus non-PcG domains, or between *control* versus *PcG mutant* genotypes. For example, a reporter integrated into a repressed PcG domain should be less inducible than one integrated into an active domain, and genetically removing PcG proteins should increase the induction of reporters in repressed PcG domains but not active domains (*Figure 1C*).

When applied to ovarian germline cells, the reporter system fulfilled these expectations. We compared hsGFP induction in differentiated nurse cells at stage 9–10 (*Figure 1D*) for an insert in an active (*Dak1*), PcG (*Antp*), or inactive (*OR67D*) region (*Figure 1E*). The reporter integrated near *Dak1* was strongly induced in nurse cells following heat shock, while the reporters integrated near *Antp* or *OR67D* did not induce following heat shock. A key test was the effect of removing *Enhancer of zest*, *E(z)*, function, the gene encoding the PRC2 H3K27 methyltransferase that is essential for Polycomb repression. **G**erm**L**ine-specific RNAi **K**nock **D**own (GLKD) of *E(z)* (*E(z)*$^{GLKD}$) relieved repression of the reporter near *Antp* and *OR67D* but had no effect at *Dak1*. Based on these results, we

**Table 1.** Annotated PcG domains larger than 10 kb in S2 cells (*Kharchenko et al., 2011*), or PcG domains in nurse or follicle cells, resembling an active domain in another cell type.

Some domains, for example NCFC2, differ in size among the cell types.

| Name | dm6 Coordinates | Associated genes | Length (kb) | Nurse cell state | Follicle cell state | S2 cell state |
|------|-----------------|------------------|-------------|------------------|---------------------|---------------|
| NC1 | 2L:2198000..2209000 | CG31668, CG33124 | 11 | PcG | Inactive | Active |
| NC2 | 2R:22673000..22692000 | ppa | 19 | PcG | Inactive | Active |
| NC3 | 3L:17950000..18070000 | Eip75B | 120 | PcG | Active/inactive | Active |
| NC4 | 3R:18415000..18455000 | fru | 40 | PcG | Active | Active |
| NC5 | 3R:18910000..18927000 | Xrp1 | 17 | PcG | Active | Active |
| NCFC1 | 3R:15985000..16041000 | srp, GATAe, pnr | 56 | PcG | PcG | Active |
| NCFC2 | 3R:30750000..30782000(NC), 30787000(FC) | zfh1 | 32(NC), 37(FC) | PcG/active | PcG | Active |
| FC1 | X:5048000..5070000 | ovo | 22 | Active | PcG | Active |
| FC2 | X:5312000..5317000 | dhd, CG4198, CG15930 | 5 | Active/inactive | PcG | Active/inactive |
| FC3 | X:19532000..19559000 | CG32532 | 27 | Inactive | PcG | Active |
| FC4 | 3L:13390000..13411000 | sens | 21 | Inactive | PcG | Inactive |
| FC5 | 3R:23286000..23314000 | pnt | 28 | Inactive | PcG | Active |
| S2FC1 | 2R:15658000..15676000 | fus | 18 | Active/inactive | PcG | Active/PcG |
| S2NC1 | 2L:6531500..6553500 | eya | 22 | Inactive | Active | Active |
| S2NC3 | 2R:14307000..14322000 | shroom | 15 | Inactive | Active | PcG |
| S2NC4 | 2R:20092500..20117500 | 18 w | 25 | Inactive | Active | PcG |
| S2NC2 | 2R:23566000..23580000 | apt | 14 | Inactive | Active | PcG |
| S2NC5 | 3L:607500..628000 | CG4337 | 20.5 | Inactive | Active | PcG |
| S2NC6 | 3R:31689500..31712000 | ttk | 22.5 | Inactive | Active | PcG |

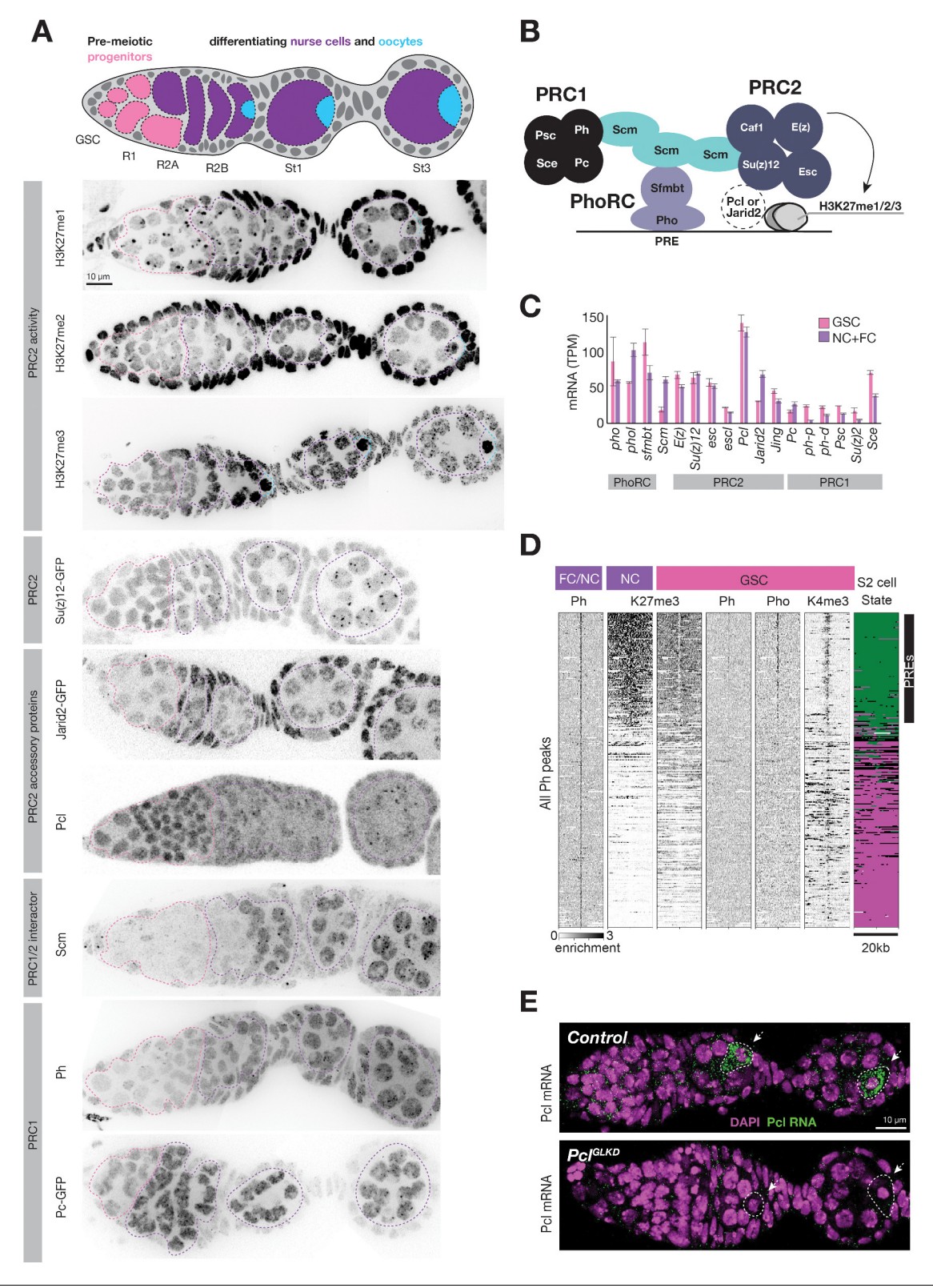

**Figure 3.** Regulation of PcG genes and activities during germline development. (**A**) Illustration highlighting stages of *Drosophila* germline differentiation and immunofluorescence staining (IF) or native GFP fluorescence of the indicated methylated H3K27 epitopes or PcG proteins. Pink dashes surround premeiotic progenitors in region 1, including germline stem cells situated at the anterior (left). Purple dashes surround nurse cells differentiating from oocytes starting in region 2. Blue dashes surround oocytes, which are located at the posterior of each follicle and are not always

*Figure 3 continued on next page*

*Figure 3 continued*

included in the optical section of ovarian tissue displayed. (B) Model showing subunits of PRC1, PRC2, PhoRC, and Scm, a putative bridge between the complexes. (C) PcG gene mRNA levels (TPM) measured by RNAseq analysis of FACS-purified GSC (pink) vs whole ovary tissue enriched in differentiated nurse and follicle cells (purple, NC+FC). (D) ChIPseq raw read depth heatmap comparing PcG proteins or histone modifications in 20 kb regions surrounding every Ph peak found in differentiated ovary tissue. In GSCs, Ph peaks in PcG domains (upper region) are associated with Pho and a 'bivalent' enrichment of H3K27me3 and H3K4me3. (E) In situ hybridization shows that *Pcl* mRNA (green) accumulates in oocytes (arrowheads, white outline). *Pcl$^{GLKD}$* serves as a control. Scale bars: D,E 10μ.

The online version of this article includes the following figure supplement(s) for figure 3:

**Figure supplement 1.** Su(z)12-GFP foci require Pcl and colocalize with Pho.

measured reporter induction in 11 additional PcG domains, three additional inactive domains, and two additional control active domains. In all cases, reporters in active domains were highly induced while reporters in PcG and inactive domains were nearly uninducible in stage 9–10 nurse cells (*Figure 1G*). These results strongly supported the utility of inserted hsGFP reporters as sensors of local chromatin in multiple types of genomic chromatin.

## PcG silencing arises during nurse cell differentiation

To address when Polycomb repression begins during nurse cell development, we measured GFP induction in reporters in active, inactive, and PcG domains during ovarian follicle development, including two stages before (*Figure 1F–H*, pink bar on x-axis) and 10 stages during and after nurse cell differentiation (*Figure 1F–H*, purple bar on x-axis). We plotted reporter induction as a function of stage, and overlayed plots from multiple reporter types to compare trends in reporter induction between insertions in different chromatin types. We observed very little difference in reporter induction between active and repressed loci in germ cells prior to the onset of meiosis in region 2A (*Figure 1G*). However, by stage 1–2, reporters in all inactive and some PcG domains were less inducible than those in active domains (*Figure 1G*). We additionally studied each reporter in an *E(z)$^{GLKD}$* genetic background to determine whether its silencing was E(z)-dependent throughout development (*Figure 1H*). While reporters in active chromatin were not affected by *E(z)$^{GLKD}$* at any stage, we reliably detected E(z)-dependent reporter silencing in all inactive and most PcG domains by stage 1–2, and in all PcG domains by stage 6. The difference between *E(z)$^{GLKD}$* and control reporter induction in inactive- and PcG-localized reporters generally increased as nurse cell development progressed, indicating that E(z)-dependent silencing strengthens over time (*Figure 1H*). We conclude that PRC2-dependent reporter gene silencing is absent in germline progenitors, initiates in both inactive and PcG domains in nurse cells after germline progenitors differentiate into nurse cells or oocytes, and strengthens as nurse cells grow.

## PRC2-mediated repression is associated with H3K27me3 modification of canonical polycomb domains in nurse cells

To further investigate the role of PRC2-mediated chromatin modification in nurse cell repression, we mapped the genomic distribution of PRC2 and its H3K27me products by ChIPseq across the genome during ovarian follicle development. We used nuclear fractionation and a fluorescence activated cell sorter to purify fixed, GFP or Tomato-labeled nuclei from a variety of ovarian cell types. We used a germline-driven NLS-GFP and G1 (2c) DAPI content as tags to purify GSCs from ovaries where germline stem cell differentiation is blocked by the *bag of marbles (bam)* mutation (*Kai et al., 2005*; *McKearin and Spradling, 1990*). (*Figure 2A*). For nurse cells, we used germline-driven tdTomato and DAPI to purify individual nurse or follicle cell stages beginning at 2c and ending at 512c DNA content (*Figure 2B*). We additionally purified tomato negative follicle cells as a somatic cell control. These methods allowed us to obtain large quantities of highly purified nuclei corresponding to every developmental stage and cell type in *Figure 1F*.

If the process giving rise to E(z)-dependent reporter repression in *Figure 1* is canonical Polycomb repression, then high levels of H3K27me3 should accumulate on Polycomb domains coincident with repression. We analyzed chromatin from purified nurse cell nuclei from late stages well after repression was complete (512c, stage 9–10), and found that H3K27me3 was highly enriched on PcG domains. Moreover, lower but significant H3K27me3 enrichment occurred on Inactive domains,

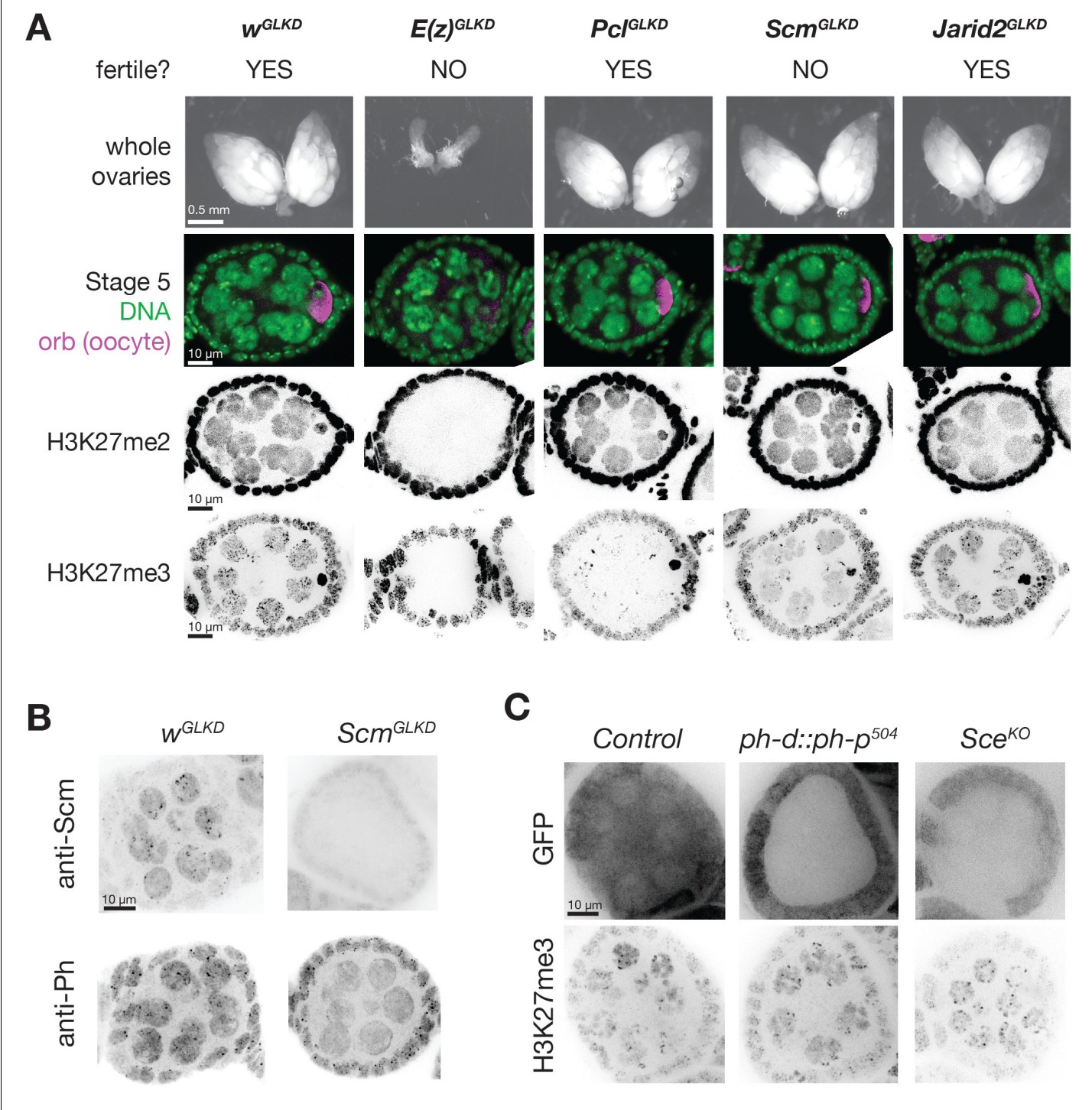

**Figure 4.** *PcG^GLKD* effects on fertility, ovarian development, and bulk H3K27 methylation. (**A**) Whole ovaries (row 1) or immunofluorescence (IF) images of stage 5 follicles antibody stained for the indicated protein epitopes or DNA (DAPI) (rows 2–5). *E(z)^GLKD* blocks oocyte differentiation (row2), and abolishes H3K27me2 and H3K27me3 staining, while *Pcl^GLKD* and *Scm^GLKD* reduce H3K27me3 but not H3K27me2. (**B**) IF images showing *Scm^GLKD* effectively removes Scm protein from nurse cells and prevents the coalescence of Ph into discreet foci. (**C**) IF images of H3K27me3 in *ph-d/ph-p* or *Sce* null mutant clones generated by mitotic recombination and visualized by lack of GFP (clonal marker) fluorescence.

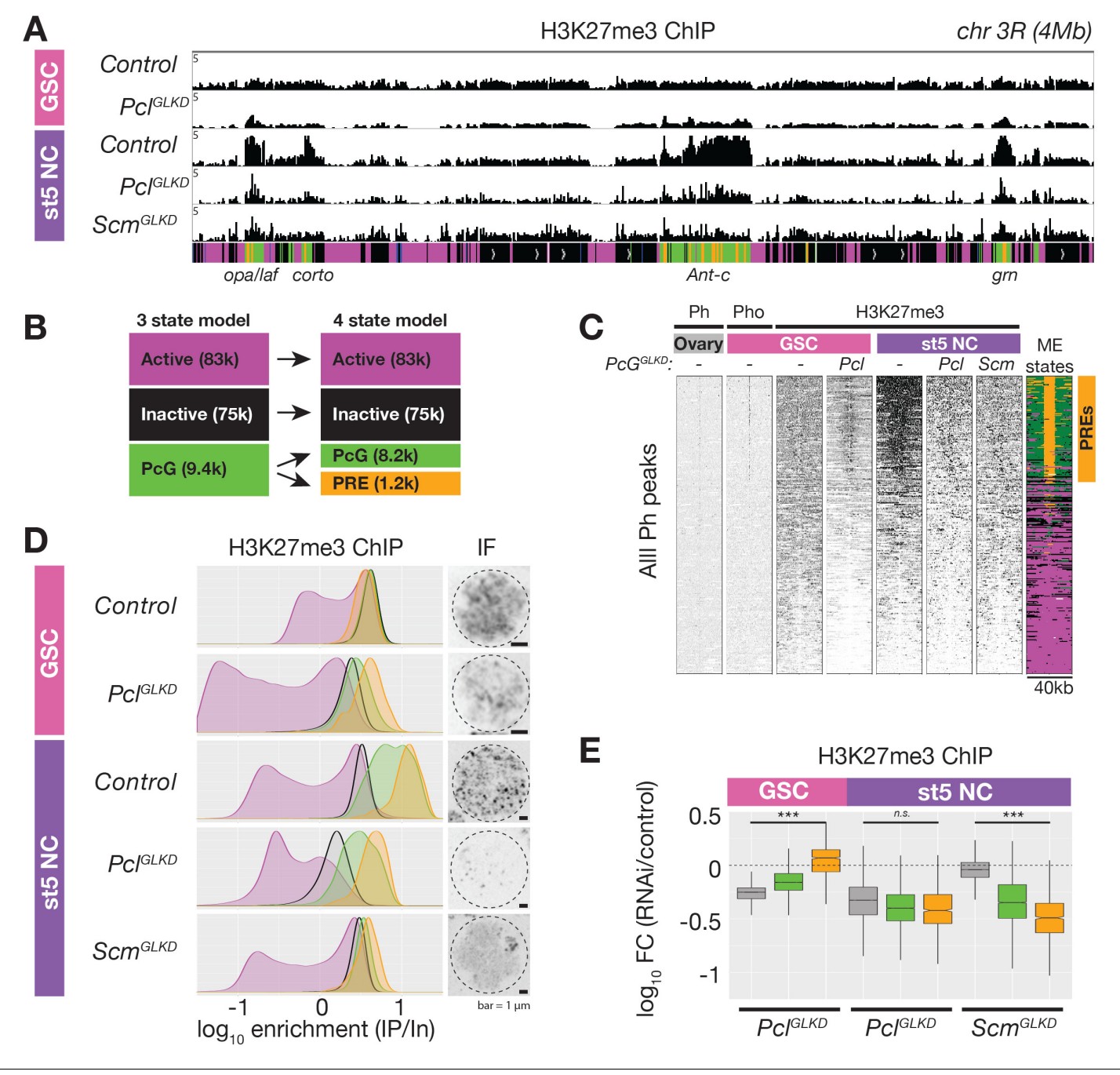

**Figure 5.** ChIPseq of *Pcl*<sup>GLKD</sup> and *Scm*<sup>GLKD</sup>. (A) Spike-in normalized H3K27me3 ChIP from FACS-purified GSC or stage 5 nurse cell (St5 NC) nuclei of the indicated genotypes in a 4 Mb region including *Ant-c*. *Pcl*<sup>GLKD</sup> in GSCs specifically depletes H3K27me3 from inactive loci. In NCs, *Pcl*<sup>GLKD</sup> and *Scm*<sup>GLKD</sup> deplete H3K27me3 from PcG loci. (B) Subdivision of 9400 PcG bins into 1200 PRE-containing bins (orange), and 8,200 PRElacking bins (green). (C) ChIPseq raw read depth heatmap showing the effect of *Pcl* and *Scm* knockdown on H3K27me3 enrichment near all ovary Ph peaks. Note that H3K27me3 enrichment on GSC PREs is revealed by *Pcl*<sup>GLKD</sup>. (D) Smoothed histograms for each indicated genotype and stage showing spike-in normalized H3K27me3 enrichment (IP/Input) in 5 kb active (magenta), inactive (black), PcG (green), and PRE-containing (orange) bins tiling the genome. IF images of H3K27me3 in GSC and St5 NC show the relationship between H3K27me3 enrichment in ChIPseq (left) and whole mount staining (right). (E) Boxplots summarizing the fold changes in H3K27me3 enrichment in inactive (grey), PcG (green), and PRE-containing (orange) 5 kb bins induced by *Pcl*<sup>GLKD</sup> or *Scm*<sup>GLKD</sup> in the indicated stages.

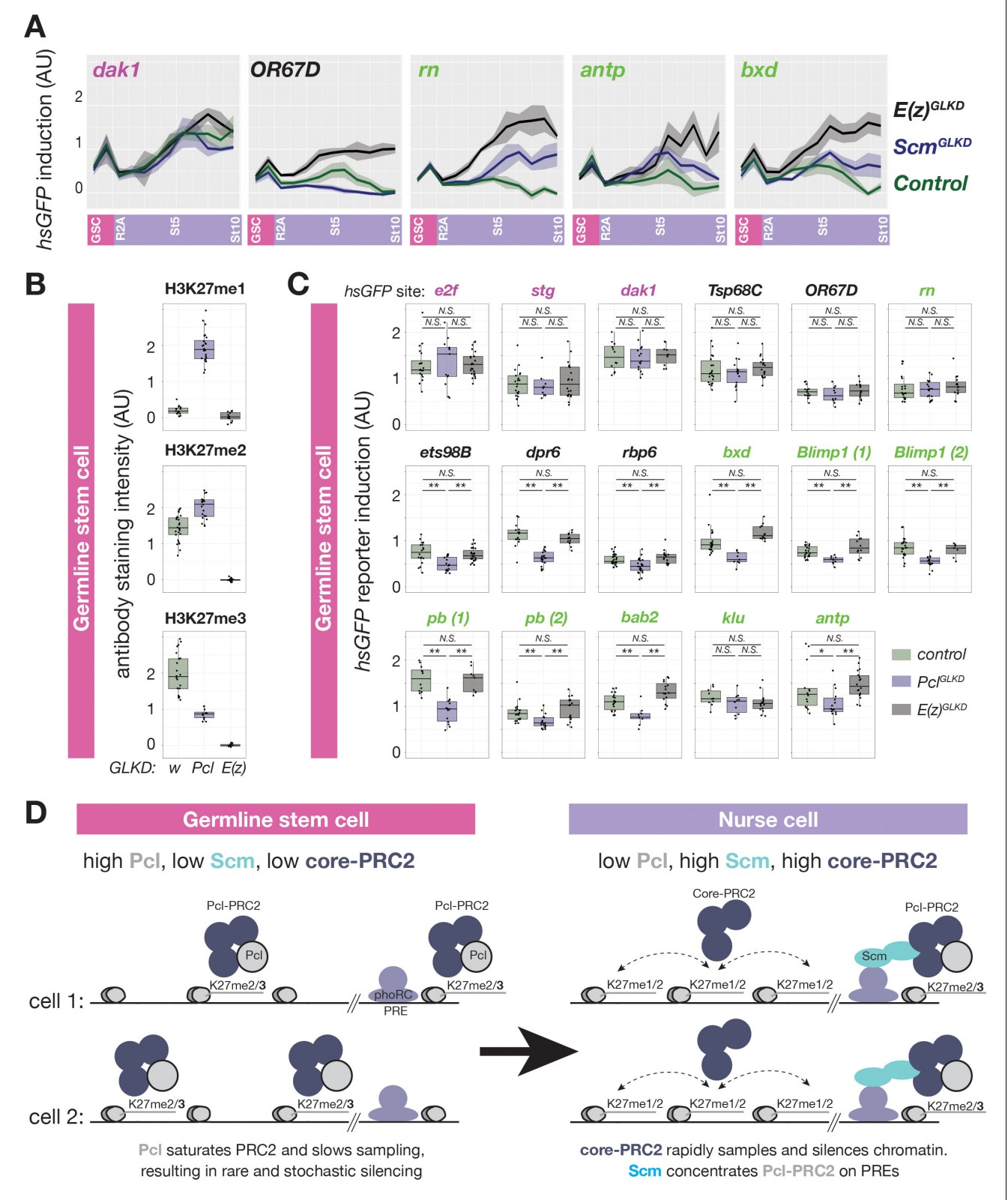

**Figure 6.** Regulation of PRC2-depenent silencing through Pcl and Scm. (**A**) The effects of *E(z)*<sup>GLKD</sup> or *Scm*<sup>GLKD</sup> versus control (*w*<sup>GLKD</sup>) on *hsGFP* reporters near an active domain (*dak1*), inactive domain (*OR67D*), or PcG domains (*rn*, *antp*, or *bxd*) throughout the indicated stages of germ cell development. Solid line indicates mean fluorescence; shading shows one standard deviation from the mean. (**B**) Quantification of relative H3K27me1/2/3 antibody staining intensity in the euchromatin of control *w*<sup>GLKD</sup>, *Pcl*<sup>GLKD</sup>, or *E(z)*<sup>GLKD</sup> GSCs. (**C**) Quantification of reporter gene induction in GSCs in

*Figure 6 continued on next page*

*Figure 6 continued*

control *w*$^{GLKD}$, *Pcl*$^{GLKD}$, or *E(z)*$^{GLKD}$. Note that *Pcl*$^{GLKD}$ reduces the induction of some inactive (black) and PcG (green) localized reporters but not active (magenta) localized reporters (*=p < 0.05, **=p < 0.01, N.S. = not significant; Student's t-test, unpaired, 2-tailed). (**D**) Sampling model for the developmental control of silencing. In GSCs, most PRC2 is associated with Pcl, whose affinity for DNA prevents PRC2 from sampling many sites, resulting in infrequent and stochastic silencing. As Pcl levels drop during differentiation, core-PRC2, having a lower affinity for DNA, is freed to sample and silence more sites. Additionally Scm is induced and concentrated on PREs, where it preferentially concentrates residual Pcl-PRC2 through cooperativity between the Scm-PRC2 interaction and Pcl-DNA interaction.

The online version of this article includes the following source data and figure supplement(s) for figure 6:

**Source data 1.** Immunofluorescence intensity measurements for *Figure 6B*.
**Source data 2.** Fluorescence intensity measurement for *hsGFP* reporters in *Figure 6C*.
**Figure supplement 1.** H3K27 methylation staining in germline progenitors IF images of ovaries of the indicated genotype stained for the indicated H3K27me epitope.

which were co-enriched for H3K27me2 (*Figure 2C,D*). Predominantly mitotic (2c) or differentiated (16c) somatic follicle cell nuclei contained a similar H3K27me3 distribution as nurse cells (*Figure 2C*). When we measured H3K27me3 enrichment on all 118 PcG domains from *Kharchenko et al., 2011*, 117 and 112 were highly enriched for H3K27me3 in nurse cells and follicle cells, respectively (*Figure 2E,F*). Genes located within PcG domains that were depleted for H3K27me3 in nurse or follicle cells (*fusilli, eyes absent, tramtrack, 18 wheeler, and apontic*) had known expression and functions in oogenesis (*Figure 2E*, *Table 1*; *Bai and Montell, 2002*; *French et al., 2003*; *Kleve et al., 2006*; *Lie and Macdonald, 1999*; *Starz-Gaiano et al., 2008*; *Wakabayashi-Ito et al., 2001*). Through manual curating, we also found that nurse cells and follicle cells each contained seven additional H3K27me3-enriched domains (two shared) that were annotated as either active domains (6) or inactive domains (1) in S2 cells (*Table 1*). In total, 111 out of 130 PcG domains (85%) were shared between all three cell types (*Figure 2F*). We conclude that nurse cells contain a highly similar collection of H3K27me3-enriched PcG domains as somatic cells, and that only a few PcG domains differ between nurse cells and other somatic cells.

## Germline stem cells lack reporter repression and H3K27me3 enrichment

An important question for understanding germ cell biology and nurse cell differentiation concerns the chromatin state of germ cell progenitors, which in the *Drosophila* ovary comprise GSCs. GSCs derive from pole cells in the embryonic gonad that become incorporated into a stem cell niche during late larval stages that represses further development via BMP signaling (reviewed in *Losick et al., 2011*). We observed a 'non-canonical' H3K27me3 distribution in *Drosophila* GSCs that is similar to that observed in early embryos (*Figure 2C*). In contrast to the canonical H3K27me3 distribution in nurse cells, GSCs did not highly enrich H3K27me3 on PcG domains. Relative to spike-in, GSCs had similar amounts of total H3K27me3 signal as nurse cells or follicle cells, but the signal was more evenly distributed across the genome (*Figure 2C,D*). We divided the genome into overlapping 5 kb bins and classified the bins according to their S2 cell chromatin state. In GSCs, we noted no greater enrichment of H3K27me3 on PcG domains compared to generally inactive domains in contrast to an approximately 10-fold enrichment in late stage nurse cells or follicle cells (*Figure 2H*). H3K27me3 signal in GSCs was depleted from S2 cell active domains (*Figure 2H*), or more specifically, genes with H3K27ac, a transcription-promoting modification, near their transcription start sites (TSSs) (*Figure 2G*). Using published data from *Li et al., 2014*, we noted a similar depletion of H3K27me3 enrichment from active domains in early embryos and a non-canonical H3K27me3 distribution in cycle 13 embryos that transforms into a canonical distribution in cycle 14 (*Figure 2I*). These experiments show that the process of nurse cell differentiation resembles somatic cell differentiation in early embryos and includes changes in the production of H3K27me3 that transform its broad distribution across non-expressed chromatin into a highly focused pattern on common PcG domains. However, in the germline, these changes occur in a single lineage of cells that are more accessible for experimentation than embryonic cell lineages.

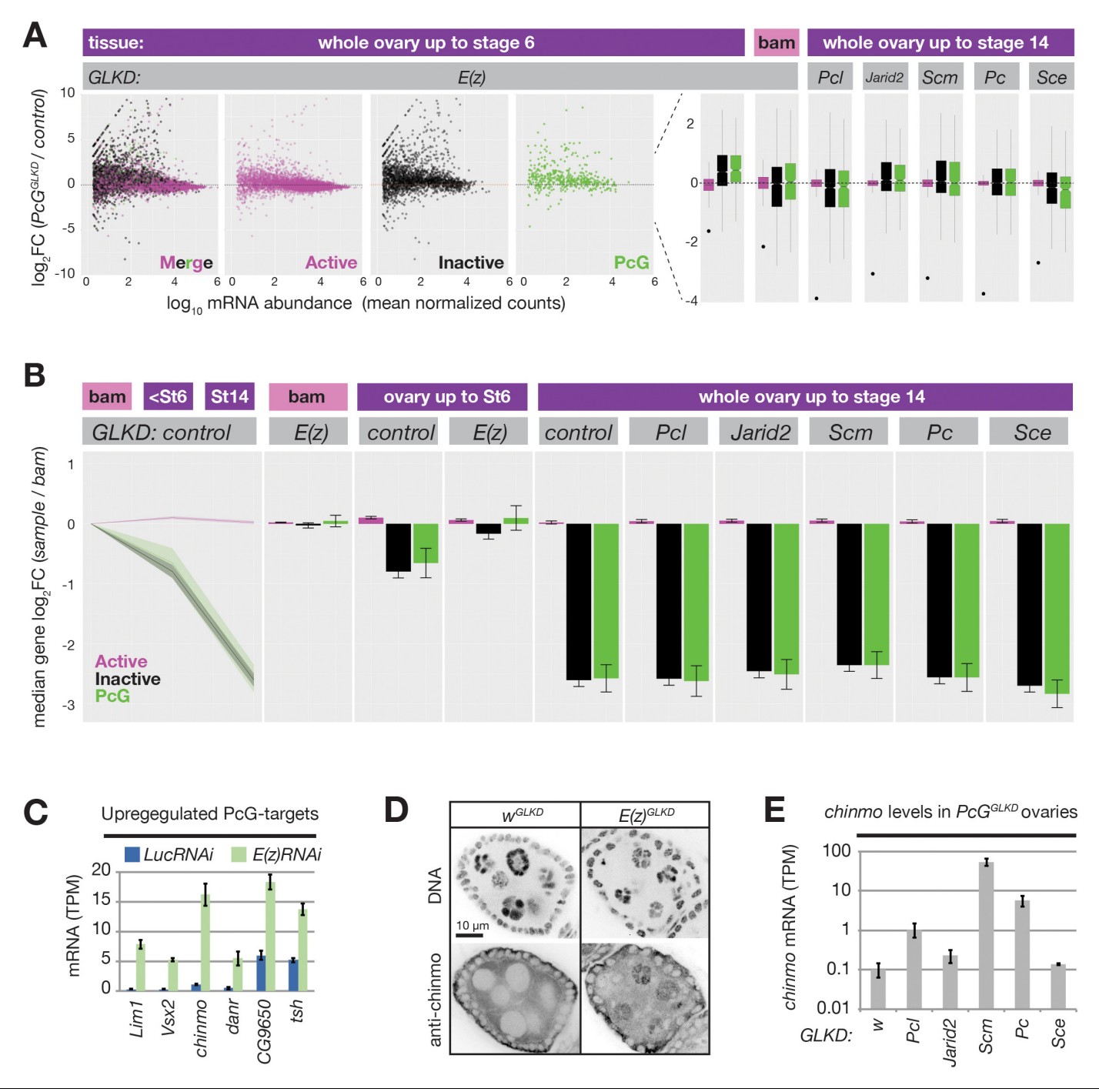

**Figure 7.** PRC2 represses endogenous genes in inactive and PcG domains in nurse cells. (A) Whole ovary RNAseq showing how indicated *PcG^GLKDs* affect gene expression in whole ovaries that fully developed (right), developed until Stage 6 (left), or failed to differentiate due to the *bam* mutation (middle). On the scatter plot (left), each dot represents a protein-coding gene colored to match the chromatin domain it resides in. Notched boxplots (right) summarize the fold-change distribution for all genes residing in each domain class. Notches show 95% confidence interval (CI) of the median, boxes show interquartile range. Black dots indicate the fold change of the PcG gene targeted by RNAi in each sample. (B) Whole ovary RNAseq comparing gene expression in each indicated stage and genotype to control, undifferentiated, *bam* mutant ovaries. (Left panel) The relative median fold changes (solid lines, 95% CI is shaded) between each class of gene (colored by domain type) plotted as a function of ovary development. (Right nine panels) The effects of various *PcG^GLKDs* or controls on relative median fold change for each class of gene. Error bars represent 95% CI of the median. (C) Mean transcripts per million (TPM) of transcription factors located in PcG domains with the highest expression (>5 TPM in *E(z)^GLKD*) and upregulation (>2.5-fold change) in *E(z)^GLKD* compared to control *Luc^GLKD*. Error bars represent one standard deviation from mean. (D) Stage 5 follicle IF

*Figure 7 continued on next page*

*Figure 7 continued*

showing Chinmo protein upregulation in *E(z)*^GLKD^ nurse cell nuclei. DNA = DAPI. (**E**) Whole ovary. RNAseq showing that GLKD of some *PcG* genes upregulate *chinmo*, compared to control (**w**). Size bars: A (row 1) 0.5 mm, (rows 2–4) 10 μm, (row 5) 1 μm.

The online version of this article includes the following figure supplement(s) for figure 7:

**Figure supplement 1.** *Scm*^GLKD^ mothers produce defective embryos.

## Early nurse cell differentiation is associated with genome-wide remodeling of PRC2 activity

To understand how nurse cells alter the pattern of PRC2 activity and initiate Polycomb silencing, we surveyed changes in the abundance and localization of PRC2 and known interacting proteins as well as all three of its H3K27 methylation products during nurse cell differentiation. Su(z)12-GFP, a core PRC2 component tagged at its endogenous locus (*Seller et al., 2019*), was evenly distributed throughout the nuclei of GSCs and nurse cell precursors, but coalesced into perinuclear foci upon nurse cell differentiation in region 2 (*Figure 3A*). Su(z)12-GFP foci persisted throughout nurse cell development, and likely corresponded to PcG domains since they colocalized with Pho (*Figure 3—figure supplement 1A*). H3K27me3 antibody staining followed a similar pattern as Su(z)12-GFP. In GSCs and nurse cell precursors, H3K27me3 antibodies broadly stained euchromatin, while in nurse cells, they stained concentrated foci in addition to a lower euchromatin background (*Figure 3A*). Our H3K27me3 ChIP experiments suggest that the concentrated H3K27me3 foci coat PcG domains while the broad euchromatin signal covers inactive chromatin. The diffuse H3K27me3 signal in region 1 germ cells increased in intensity in region 2 as nurse cells and oocytes begin to differentiate during a prolonged pre-meiotic Sphase and prophase. Oocytes (*Figure 3A*, blue outline) retained the intense, diffuse signal as they slowly progressed through prophase, while nurse cells (*Figure 3A*, purple outline) depleted it as they exited arrest and underwent repeated rounds of DNA replication. Antibodies detecting H3K27me1 and me2 stained 1 or two heterochromatic foci per nucleus, but also generally stained euchromatin (*Figure 3A*), consistent with their enrichment throughout euchromatin in ChIP experiments (*Figure 2D*, *Lee et al., 2015*; *Wang et al., 2018*). However, germ cells had significantly less H3K27me1 and me2 signals than somatic cells, and germline stem cells and region 1 cysts had the lowest levels of H3K27me1 in any ovarian cell type. We conclude that nurse cells alter PRC2 activity in two ways as they differentiate in region 2. First, nurse cells concentrate PRC2 and H3K27me3 on PcG domains, and second, nurse cells increase the levels of H3K27me1 throughout euchromatin.

## Identification of Pcl and Scm as candidate regulators of polycomb initiation

We hypothesized that PRC2-interacting proteins regulate the remodeling of PRC2 activity upon nurse cell differentiation. For example, nurse cells could induce PhoRC, which binds PREs within PcG domains, Scm, a putative link between PhoRC and other PcG complexes, or the PRC2 accessory proteins Pcl and Jarid2, (*Figure 3B*) to trigger PRC2 concentration on PcG domains. We compared the mRNA abundance of various PRC2-interacting proteins in FACS-purified GSCs and a mixed whole ovary sample comprised mostly of differentiated nurse and follicle cells. While mRNAs encoding PhoRC were similarly abundant in GSCs and differentiated cells, *Scm* mRNA was threefold more enriched in differentiated cells than GSCs (*Figure 3C*). By antibody staining, Scm protein was nearly undetectable in GSCs and mitotic cells in region 1 but was highly induced upon nurse cell differentiation in region 2 (*Figure 3A*). In contrast, Pho protein was expressed and enriched on PREs in GSCs, but was apparently unable to concentrate Ph or high levels of H3K27me3 (*Figure 3D*). Therefore, Scm, but not PhoRC, induction correlates with the enrichment of PRC1 and PRC2 on PcG domains as nurse cells differentiate in region 2.

In contrast to Scm and other PcG-proteins, Pcl protein was uniquely highly abundant in GSCs and mitotic cells in region 1 and was depleted from differentiating nurse cells in region 2 (*Figure 3A*). By stage 1, the small amount of remaining Pcl was mostly concentrated onto PcG domains. Although *Pcl* mRNA abundance in GSCs and differentiated stages was similar (*Figure 3C*), *Pcl* mRNA was asymmetrically localized in each follicle, with nurse cells accumulating much lower levels of *Pcl* mRNA than oocytes (*Figure 3E*). Jarid-2, which competes with Pcl for binding to PRC2, remained at

a low constant level throughout germ cell development compared to higher levels in follicle cells (*Figure 3A*). We conclude that out of multiple PRC2 interacting proteins, nurse cells uniquely regulate Pcl and Scm levels to potentially remodel PRC2 activity and induce Polycomb silencing.

## Functions of Pcl and Scm in the formation of PcG domains in nurse cells

To understand how Pcl and Scm regulate PRC2 activity and silencing during nurse cell differentiation, we depleted Pcl and Scm from germ cells using germline-specific RNAi knockdown. We confirmed that our RNAi constructs efficiently depleted both genes using multiple approaches, including antibody staining, in situ hybridization, and RNA sequencing (*Figure 3E*, *Figure 4B*). In contrast to $E(z)^{GLKD}$, which induced follicles to degenerate and mislocalize the oocyte marker, Orb, at stage 5, $Pcl^{GLKD}$ and $Scm^{GLKD}$ produced mature eggs (*Figure 4A*). However, the mature eggs produced by $Scm^{GLKD}$ females failed to hatch after fertilization. At stage 5, $E(z)^{GLKD}$ completely eliminated both H3K27me2 and me3 staining, while $Pcl^{GLKD}$ and $Scm^{GLKD}$ had no effect on H3K27me2 levels. However, $Pcl^{GLKD}$ reduced the intensity of both the punctate and diffuse H3K27me3 signals in stage 5 nurse cells. Measured by spike-in normalized ChIPseq in stage 5 nurse cells, $Pcl^{GLKD}$ reduced H3K27me3 enrichment throughout inactive and PcG domains (*Figure 5A*). To look for changes that may specifically occur at PRE sites, we subdivided PcG domains into 5 kb bins either containing (*Figure 5B*, orange), or lacking (*Figure 5B*, green) a PRE. While PRE-containing bins had higher H3K27me3 enrichment than non-PRE containing bins in nurse cell PcG domains (*Figure 5D*), $Pcl^{GLKD}$ equivalently depleted H3K27me3 from both types of PcG bins as well as inactive bins (*Figure 5E*). In addition, $Pcl^{GLKD}$ inhibited Su(z)12-GFP from concentrating into large perinuclear foci (*Figure 3—figure supplement 1B*). Together, these data show that Pcl generally promotes H3K27me3 production by PRC2 and is required to fully concentrate PRC2 onto PcG domains.

While $Pcl^{GLKD}$ generally depleted H3K27me3 signal throughout the genome, $Scm^{GLKD}$ specifically depleted H3K27me3 signal from PcG domains. Compared to controls, $Scm^{GLKD}$ eliminated many prominent H3K27me3 foci, but not the general euchromatic staining (*Figure 4A*, *Figure 5D*). By spike-in normalized ChIPseq, $Scm^{GLKD}$ specifically depleted H3K27me3 from PcG but not inactive bins, and PRE-containing bins were the most depleted bin type (*Figure 5E*). Therefore, Scm is required to enrich PRC2 activity on PREs, and $Scm^{GLKD}$ nurse cells have a similar non-canonical H3K27me3 distribution as GSCs, which normally lack Scm (*Figure 5D*).

Consistent with promoting PRC2 activity at PcG domains, Scm localized to PcG domain foci in nurse cells. $Scm^{GLKD}$ not only abolished H3K27me3 enrichment into these foci, but also enrichment of the PRC1 component, Ph (*Figure 4B*). Because Scm had previously been shown to directly associate with Ph, we wondered whether Scm recruits PRC2 to PcG domains through Ph. We generated mutant clones disrupting PRC1 components, Ph or Sce, but did not observe alterations to the distribution of H3K27me3 in nurse cells (*Figure 4C*). Therefore, Scm concentrates the majority of PRC2 activity on PcG domains without PRC1.

Together, our experiments show that both Pcl and Scm are required to concentrate H3K27me3 on nurse cell PcG domains. However, while Scm promoted high levels of H3K27me3 only on PcG domains, Pcl generally promoted H3K27me3 on all domains, suggesting that Scm but not Pcl contributes to PRC2 specificity for PcG domains. If Scm specifically acts at PcG domains, it should also only promote silencing at PcG domains but not inactive domains. Indeed, $Scm^{GLKD}$ partially disrupted the silencing of PcG domain reporters near *rn*, *antp*, and *bxd*, but not an active domain reporter near dak1 or an inactive domain reporter near OR67D in late stage nurse cells (*Figure 6A*). However, PcG domain reporters were significantly less inducible in $Scm^{GLKD}$ than $E(z)^{GLKD}$ late stage nurse cells, suggesting that in the absence of Scm, PRC2 can still partially silence PcG domains (*Figure 6A*). Thus, while Scm induction in differentiating nurse cells can explain how nurse cells enhance PcG domain silencing by concentrating PcG proteins, it cannot explain how nurse cells initiate PRC2-depenent silencing at inactive domains or Scm-depleted PcG domains.

## Pcl inhibits inactive and PcG domain silencing in GSCs

The higher expression of Pcl in GSCs compared to nurse cells beginning in region 2b (*Figure 3A*) raised the question of what effect Pcl levels have on PRC2 activity and silencing. In contrast to nurse cells, where $Pcl^{GLKD}$ reduced H3K27me3 throughout inactive and PcG domains, in GSCs, $Pcl^{GLKD}$ preferentially reduced H3K27me3 from inactive domains but not PREs (*Figure 5C–E*). Following Pcl

knockdown in GSCs, inactive domain bins showed a 1.8-fold decrease in H3K27me3 enrichment while PRE-containing bins showed a median 1.2-fold increase in H3K27me3 enrichment (*Figure 6E*). However, the median H3K27me3 enrichment on PREcontaining bins in *Pcl^GLKD* GSCs was still three-fold lower than in wild-type stage five nurse cells.

Because our H3K27me3 ChIP experiments suggested that *Pcl^GLKD* generally depleted H3K27me3 from inactive bins in GSCs, we next measured how *Pcl^GLKD* affects the levels H3K27me1 and me2, which are normally enriched on inactive domains. We imaged ovaries stained with antibodies against all three H3K27 methylation states (FigS4) and compared the mean intensity of antibody signal in GSC euchromatin between *Pcl^GLKD* and a negative (*w^GLKD*) and positive control (*E(z)^GLKD*) (*Figure 6B*, *Figure 6—figure supplement 1*). Compared to *w^GLKD*, *Pcl^GLKD* decreased the H3K27me3 signal in GSCs by 2.3-fold compared to the 1.8-fold decrease on Inactive domains measured by spike-in ChIPseq. In contrast, *Pcl^GLKD* increased the H3K27me2 signal by 1.4-fold and increased the H3K27me1 signal by 13-fold (*Figure 6B*). Therefore, Pcl promotes H3K27me3 and inhibits H3K27me1 on inactive domains in GSCs.

Because H3K27me1 levels increase and Pcl levels decrease as nurse cells differentiate and initiate PRC2-dependent silencing, we tested whether premature Pcl depletion could ectopically activate PRC2-dependent gene silencing in GSCs. Normally, GSCs lack PRC2-dependent silencing, as shown by the lack of effect of *E(z)^GLKD* in GSCs on reporter induction (*Figure 6C*). In contrast, *Pcl^GLKD* in GSCs significantly suppressed reporter induction in 3/5 inactive, 7/9 PcG domains, but 0/3 active domains (*Figure 6C*) and did not enhance induction in any domain. While the magnitude of reporter repression in inactive and PcG domains in *Pcl^GLKD* GSCs was less than twofold for most sites, it was comparable to the magnitude of PRC2-dependent silencing observed in early stage nurse cells (*Figure 1G–H*). Thus in GSCs, Pcl inhibits, rather than promotes PRC2-depenent silencing, and developmentally programmed depletion of Pcl during nurse cell differentiation contributes to the onset of silencing throughout inactive and PcG domains.

## Role of Polycomb silencing on nurse cell development

Despite dramatically altering PRC2 activity in germ cells, both *Pcl^GLKD* and *Scm^GLKD* females produced seemingly normal eggs that initiated early embryonic development. However, eggs derived from *Scm^GLKD* nurse cells failed to hatch, and displayed a classic Polycomb defect after germ band extension where anterior segments express the most posterior Hox gene, Abd-b (*Figure 7—figure supplement 1*). In contrast, *E(z)^GLKD* nurse cells do not produce eggs because they degenerate at stage 6 (*Figure 3A*). To understand how Polycomb repression mediates oogenesis, we next determined how each knockdown affects nurse cell gene expression. First, we focused on the stages of nurse cell differentiation, by extracting and sequencing mRNA from control and *E(z)^GLKD* ovaries containing equivalent stage distributions up to stage 6. We then used DEseq2 (*Love et al., 2014*) to quantify the effect of *E(z)^GLKD* on protein-coding gene expression, as a function of resident chromatin types. In ovaries containing follicles up to stage 6, *E(z)^GLKD* upregulated the majority of genes in both inactive and PcG domains by a median 1.4-fold. In GSCs, where *E(z)^GLKD* has no effect on reporter induction, we observed no global effect on gene expression in any domain type (*Figure 7A*).

We additionally examined how E(z) contributes to gene expression changes that accompany differentiation. We used DEseq2 to compare *bam* mutant ovaries containing GSCs to either wild-type ovaries containing nurse cells up to stage 6 or wild-type ovaries containing nurse cells up to stage 14 (*Figure 7B*). We observed a striking trend where mRNA from many inactive and PcG domains decreased relative to active domains during nurse cell differentiation.

By stage 6, the median fold change of inactive and PcG genes was nearly twofold less than active domains, and by stage 14, this difference grew to sixfold. *E(z)^GLKD* almost completely suppressed this effect, arguing that the growing disparity between active and inactive/PcG gene expression is largely due to PRC2-induced repression in inactive/PcG domains and not developmentally-enhanced activation in active domains (*Figure 7B*). Unlike *E(z)^GLKD*, germline knockdown of *Pcl*, *Scm*, *Pc*, *Sce*, or *Jarid2* did not cause widespread gene expression changes in nurse cells and did not prevent the developmentally induced repression of genes in inactive and PcG domains (*Figure 7A,B*).

We hypothesized that PcG proteins in addition to core PRC2, including Scm and Pcl, enhance silencing at relatively rare PcG domains that are particularly vulnerable to activation in nurse cells. To find these vulnerable loci, we identified the most highly upregulated PcG domain-localized genes in

$E(z)^{GLKD}$. *chinmo*, a Jak/Stat signaling target that is normally expressed in both male and female germline stem cells (*Flaherty et al., 2010*; *Grmai et al., 2018*; *Ma et al., 2016*), was the most highly induced transcription factor following $E(z)^{GLKD}$ in ovaries (*Figure 7C*). We used anti-Chinmo antibodies to confirm that $E(z)^{GLKD}$ induced high Chinmo levels in nurse cells but not follicle cell nuclei (*Figure 7D*). Finally, we tested whether other PcG proteins are required for *chinmo* silencing. Compared to control $w^{GLKD}$, we observed *chinmo* upregulation in $Pcl^{GLKD}$, $Pc^{GLKD}$, and $Scm^{GLKD}$, and *chinmo* was most highly upregulated in $Scm^{GLKD}$, which blocks both PRC1 and PRC2 recruitment to PcG domains (*Figure 7E*). Together with our reporter results, these RNAseq experiments show that while Scm and other PcG proteins enhance Polycomb silencing at PcG domains, they are not required to silence the majority of genes during nurse cell differentiation.

## Discussion

### The *Drosophila* germline is a powerful system for studying chromatin regulation

The work described here shows that the *Drosophila* female germline has multiple advantages for studying the developmental regulation of chromatin silencing both before and during differentiation. Female GSCs continuously divide to produce new undifferentiated progenitors, which expand and differentiate into nurse cells or oocytes, generating large amounts of a much simpler tissue than a developing embryo. Additionally, we developed an inducible reporter assay compatible with the female germline that sensitively responds to developmental changes in local chromatin repression in individual cells. In contrast to RNAseq, which measures steady state RNA levels, or ChIPseq, which correlates chromatin epitopes with their perceived function on gene expression, our reporters directly test how local chromatin influences the inducibility of surrounding genes, and are easily combined with tissue specific knockdowns to identify trans-acting factors contributing to reporter inducibility. Finally, our genetic engineering approach allows any construct (not just hsGFP) to be efficiently integrated into many pre-existing 'donor' sites, including those used previously with other reporters, or sites heavily silenced by repressive chromatin in differentiated cells. Although the number of different donor sites in certain types of chromatin is currently limited, new sites continue to be generated using CRISPR/Cas9 targeting and the method can ultimately be applied virtually anywhere in the genome (*Kanca et al., 2019*).

### Female GSCs have non-canonical 'ground state' chromatin similar to early embryos

Our analysis of Polycomb repression with reporters, ChIP, and PcG-gene knockdowns provided numerous insights into how chromatin affects gene expression and female germline development in *Drosophila*. We found that GSCs, the precursors of oocytes and nurse cells, contain a non-canonical, binary distribution of moderate H3K27me3 enrichment on all transcriptionally inactive loci and very low enrichment on active chromatin (*Figure 2H*). We observed a similar non-canonical H3K27me3 distribution in early fly embryos (*Figure 2I*), suggesting that noncanonical chromatin represents a 'ground state' for progenitors that will propagate future generations of undifferentiated germ cells or somatic cells that differentiate into specialized tissues. Such non-canonical chromatin was first identified in mouse oocytes and preimplantation embryos (*Liu et al., 2016*; *Zheng et al., 2016*). If non-canonical H3K27me3 chromatin is a characteristic of undifferentiated, totipotent cells, what function might it confer to account for its conservation?

Our experiments confirmed previous work (*Iovino et al., 2013*) showing that chromatin modified by PRC2 is essential for the *Drosophila* female germ cell cycle. Germline cysts lacking PRC2 are unable to stably generate oocytes, and $E(z)^{GLKD}$ nurse cells mis-express multiple genes and degenerate at about stage 5 (*Figure 7AB*, *Figure 4A*). In contrast, GSCs lacking PRC2 properly populate their niche, divide, and produce daughters that interact with female follicle cells and begin nurse cell differentiation. Removing PRC2 activity from GSCs did not generally increase the steady state abundance of genes or the inducibility of reporters in H3K27me3-enriched inactive or PcG domains (*Figure 7A,B*, *Figure 1H*). These results suggest that PRC2 and non-canonical chromatin lack vital functions in undifferentiated germline progenitors but are critical for repressing genes upon differentiation. However, we cannot discount a requirement for PRC2 or non-canonical chromatin under

stress conditions or prolonged aging. For example, PRC2 could promote the long-term maintenance of female GSCs, similarly to how it maintains male germline progenitors in flies and mice (*Eun et al., 2017*; *Mu et al., 2014*).

### *Drosophila* nurse cells differentiate as somatic cells

Germline cysts and nurse cells are found in diverse animal species across the entire phylogenetic spectrum, but their function has been well studied mostly in insects such as *Drosophila* where they persist throughout most of oogenesis. While nurse cells have traditionally been considered germ cells rather than late-differentiating somatic cells, we show that that *Drosophila* nurse cells initiate Polycomb silencing and enrich PRC2 activity on a nearly identical collection of PcG domains as somatic cells (*Figure 2E,F*). In more distant species, such as mice, nurse cells initially develop in a similar manner within germline cysts and contribute their cytoplasm to oocytes, but undergo programmed cell death before the vast majority of oocyte growth (*Lei and Spradling, 2016*; *Matova and Cooley, 2001*). Consequently, it remains an open question whether somatic differentiation plays a role in nurse cell function in mammals and many other groups.

### Conserved roles for polycomb silencing in gametogenesis

The size and composition of oocyte cytoplasm are uniquely tailored to promote optimal fecundity and meet the demands of early development. In some species, including flies, nurse cells synthesize large amounts of specialized ooplasm to rapidly produce multitudes of large, pre-patterned embryos. In others, including mammals, oocytes more slowly synthesize the majority of ooplasm. Interestingly, both ooplasm synthesis strategies apparently require Polycomb silencing. However, the nurse cell-based strategy in flies primarily requires PRC2 but not PRC1 to silence hundreds of somatic genes (*Figure 7* and *Iovino et al., 2013*), while the oocyte-based strategy in mice requires PRC1 but not PRC2 (*Erhardt et al., 2003*; *Posfai et al., 2012*).

Why might this difference have arisen?

Different strategies of ooplasm synthesis may have evolved to be compatible with noncanonical germ cell chromatin. Our staining experiments show that *Drosophila* oocytes maintain a widely distributed, non-canonical H3K27me3 distribution similar to pre-meiotic precursors or mouse oocytes (*Figure 3A*), suggesting that non-canonical chromatin is conserved and maintained throughout the germ cell cycle (*Figure 1A*). Similar to mouse oocytes, *Drosophila* spermatocytes also contain non-canonical chromatin (*El-Sharnouby et al., 2017*) and autonomously synthesize large amounts of cytoplasm by deploying PRC1 but not PRC2 (*Chen et al., 2005*). Thus, three different types of germ cells are filled with large amounts of differentiated cytoplasm that requires Polycomb silencing for its synthesis, but nevertheless maintain a non-canonical, silencing-deficient PRC2 activity.

### Regulation of non-canonical chromatin in GSCs

The conservation of undifferentiated, non-canonical chromatin despite a strong selection for Polycomb silencing during ooplasm synthesis argues that non-canonical chromatin must have a presently unappreciated fundamental purpose in germ cells. Noncanonical chromatin could regulate multigenerational processes like mutation, recombination, or transposition, that are not easily assayed in sterile individuals. Tests of these ideas will require a better understanding of how non-canonical chromatin is regulated and methods to disrupt non-canonical chromatin without disrupting other functions required for germline viability. Additionally, non-canonical chromatin could simply result from the silencing- incompetent PRC2 we observed in progenitors.

We uncovered Pcl as both an inhibitor of PRC2 silencing (*Figure 6C*) and promoter of non-canonical chromatin in GSCs (*Figure 5*). $Pcl^{GLKD}$ dramatically altered the footprint of PRC2 activity in GSCs. $Pcl^{GLKD}$ favored H3K27me3 enrichment on PREs versus inactive domains (*Figure 5A–D*), and increased the total amount of H3K27me1 by 13-fold and H3K27me2 by 1.4-fold and decreased the total amount of H3K27me3 by 1.8 fold (*Figure 6B*). By binding DNA through its winged-helix domain, Pcl triples PRC2's residence time on chromatin and promotes higher states of H3K27 methylation in vitro (*Choi et al., 2017*). In GSCs, Pcl could simply change the result of each PRC2-chromatin binding event from H3K27me1 to me3. However, it is hard to imagine how an equivalent number of nucleosomes bearing a higher H3K27 methylation state could explain how Pcl inhibits silencing. Instead, we propose that Pcl inhibits silencing by reducing the number of PRC2-chomatin binding

events per unit time by increasing the residence time of PRC2 on chromatin with each binding event (*Figure 6D*). In our model, $Pcl^{GLKD}$ would not only convert many H3K27me3 nucleosomes into H3K27me1 nucleosomes, it would also convert many unmethylated nucleosomes into H3K27me1 nucleosomes. $Pcl^{GLKD}$ would more subtly affect H3K27me2 abundance because it simultaneously increases the number of PRC2-chromatin binding events while reducing the probability of each binding event leading to H3K27me2 versus me1.

By reducing the number of PRC2 binding events, Pcl could increase the abundance of unmethylated H3K27 residues available for acetylation – a transcription promoting modification (*Boija et al., 2017*). In both flies and mammals, PRC2 transiently associates with chromatin to mono- and dimethylate H3K27 outside of traditional PcG domains, blocking H3K27 acetylation and antagonizing transcription (*Ferrari et al., 2014*; *Lee et al., 2015*). We similarly found strong and widespread PRC2-dependent silencing in H3K27me1/2 enriched chromatin in nurse cells. Because inactive domain silencing was not affected by depletion of Pcl, Jarid2, or H3K27me3 (*Figure 7A,B*), we propose that core-PRC2, but not Pcl-PRC2 or H3K27me3, primarily silences inactive chromatin (*Figure 6D*).

In GSCs, abundant Pcl could saturate PRC2, effectively depleting faster-sampling core-PRC2 complexes in favor of slower sampling Pcl-PRC2. In somatic embryonic cells, Pcl is present in a small fraction of PRC2 complexes (*Nekrasov et al., 2007*). Compared to other fly tissues, Pcl mRNA is most abundant in the ovary (*Leader et al., 2018*), and within the ovary, Pcl protein is much more abundant in GSCs and nurse cell precursors than differentiated nurse cells and somatic cells (*Figure 3A*). Within each differentiated germline cyst, Pcl mRNA is depleted from nurse cells and enriched in oocytes (*Figure 3E*), suggesting that Pcl protein levels may be regulated by an mRNA transport mechanism induced in region 2 that also triggers the differentiation of oocytes from nurse cells (*Huynh and St Johnston, 2000*).

## Changes in PRC2-interacting proteins initiate nurse cell polycomb silencing

Pcl, and a second PRC2-interacting protein, Scm, regulate the transition from noncanonical to canonical chromatin and initiate Polycomb repression. During nurse cell differentiation, we propose that Pcl depletion frees core-PRC2 to rapidly sample and silence inactive domains, while Scm (which is absent from the GSC) induction recruits high levels of PRC1 and PRC2 activity around PREs (*Figure 6D*). $Scm^{GLKD}$ nurse cell chromatin retained a noncanonical H3K27me3 pattern characteristic of GSCs, as if differentiation at PcG domains had not occurred (*Figure 5D*). In mice, Scm homologue, Scml2, similarly associates with PcG domains to recruit PRC1/2 and silence PcG targets during male germline development (*Hasegawa et al., 2015*; *Maezawa et al., 2018*). However, unlike its fly orthologue in female GSCs, Scml2 is expressed in male germline precursors. This difference could explain why mammalian PGCs partially enrich PRC2 activity on CGIs (*Lesch et al., 2013*; *Zheng et al., 2016*) while fly female GSCs do not enrich PRC2 on specific sites. While PcG domain-associated Scm is sufficient to enrich PRC2 activity above background levels found throughout inactive chromatin, a second PRC2 interacting protein, Pcl, is additionally required to promote full PRC2 and H3K27me3 enrichment on PcG domains (*Figure 3—figure supplement 1*, *Figure 5*). Because Scm oligomerizes and interacts with PRC2 in vitro (*Kang et al., 2015*; *Peterson et al., 1997*), it could form an array of PRC2 binding sites anchored to PREs through Sfmbt (*Alfieri et al., 2013*; *Frey et al., 2016*). We propose that two cooperative interactions, PRC2 with PRE-tethered Scm, and Pcl with DNA, preferentially concentrate H3K27me3-generating Pcl-PRC2 versus H3K27me1/2-generating core-PRC2 on PcG domains (*Figure 6D*). H3K27me3 could then be further enriched by H3K27me3-induced allosteric PRC2 activation through the Esc subunit (*Margueron et al., 2009*).

By promoting PRC1 and PRC2 concentration on PcG domains, Scm enhances silencing on PcG-localized reporters. While *Scm*-depleted nurse cells completed oogenesis, a subset of PcG domain-localized PcG including *chinmo* and the posterior Hox gene, *Abd-b*, escaped repression and were potentially loaded into embryos. Eggs derived from $Scm^{GLKD}$ nurse cells failed to hatch, and misexpressed *Abd-b* in anterior segments following germ band elongation (*Figure 7—figure supplement 1*). This defect more closely resembled maternal plus zygotic than maternal-only *Scm* mutants (*Breen and Duncan, 1986*), suggesting that $Scm^{GLKD}$ may deplete both maternal and zygotic Scm.

 Chromosomes and Gene Expression

However, we cannot exclude the additional possibility that mis-regulation of maternal Polycomb targets like *chinmo* contribute to the subtle embryonic defects observed in maternal-only Scm mutant clones.

### Development role of germline chromatin changes

Further study of the Polycomb-mediated repression described here will help define the gene regulation program of *Drosophila* nurse cells and its contribution to oocyte growth. Additional characterization and perturbation of non-canonical chromatin throughout the germ cell cycle will yield further insights into its function in development. Finally, incorporating studies of other chromatin modifications, including H3K9me3-based repression, during germ cell development will contribute to a fuller understanding of how chromatin contributes to an immortal cell lineage.

## Materials and methods

### *Drosophila* lines used in this study

| RNAi Lines | Source | Stock number | Insertion site | Vector: | Other info: |
|---|---|---|---|---|---|
| *UASp-E(z)*<sup>RNAi</sup> | Bloomington | 36068 | attp40 | Valium22 | TRiP.GL00486 |
| *UASt-luciferase*<sup>RNAi</sup> | Bloomington | 31603 | attp2 | Valium1 | TRiP.JF01355 |
| *UASp-Scm*<sup>RNAi</sup> | Bloomington | 35389 | attp2 | Valium22 | TRiP.GL00308 |
| *UASt-Jarid2*<sup>RNAi</sup> | Bloomington | 40855 | attp40 | Valium20 | TRiP.HMS02022 |
| *UASz-Pcl*<sup>RNAi</sup> | This Study | | attp40 | UASz-miR | |
| *UASz-w*<sup>RNAi</sup> | This Study | | attp40 | UASz-miR | |
| *UASz-Sce*<sup>RNAi</sup> | This Study | | attp40 | UASz-miR | |
| *UASz-Pc*<sup>RNAi</sup> | This Study | | attp40 | UASz-miR | |
| Mutant Alleles | Source | Stock Number | | citation | expand genotype |
| *ph-d504 ph-p504* | Bloomington | 24162 | | *Dura et al., 1987* | FRT101 ph-d<sup>504</sup> ph-p<sup>504</sup> |
| *Sce*<sup>KO</sup> | Bloomington | 80157 | | *Gutiérrez et al., 2012* | FRT82B Sce<sup>KO</sup> |
| *bamΔ86* | Bloomington | 5427 | | *Bopp et al., 1993* | bamΔ86 |
| Gal4 Driver Lines | Source | Stock Number | Insertion Site | citation | expand genotype |
| *Nos-Gal4* | Bloomington | 4937 | 3R:10407270 | *Van Doren et al., 1998* | P{GAL4::VP16- nos.UTR} |
| *MTD-Gal4* | Bloomington | 31777 | X, 2, 3R:10,407,270 | *Petrella et al., 2007* | P{otuGAL4::VP16}; P{GAL4-nos.NGT}; P{GAL4::VP16nos.UTR} |
| Fluorescent Lines | Source | Stock Number | Insertion Site | citation | expand genotype |
| *UASz-tdTomato UASp-NLS-GFP* | This study Zhao Zhang | | VK33 Chr3 | | |
| *Jarid2-GFP* | Bloomington | 66754 | attP40 | *Kudron et al., 2018* | P{Jarid2-GFP.FPTB} |
| *Su(z)12-sfGFP* | Pat O'Farrell | | Su(z)12 | *Seller et al., 2019* | |
| *Pc-GFP* | Bloomington | 9593 | Chr3 | *Dietzel et al., 1999* | P{PcT:Avic\GFP-EGFP} |
| *vasa-GFP; bam*<sup>Δ86</sup> | Spradling Lab | | Chr3 | *Kai et al., 2005* | |
| *hsGFP donor* | This study | | Chr2 | | P{FRT-attB-hsGFP- attB-FRT} |
| *His2Av-RFP* | Bloomington | 23650 | 71B | Stefan Heidmann | P2His2Av-mRFP1}III.1 |
| hsGFP reporters | Source | Original Bloomington # | Replaced MiMIC (orientation) | | Release6 Coordinates |
| Active Domains | | | | | |
| *hsGFP::Dak1* | This study | BL33087 | MI00814 (F) | | 3R:26,054,482 |
| *hsGFP::Stg* | This study | BL38149 | MI04651 (R) | | 3R:29,259,877 |

*Continued on next page*

 

*Continued*

| RNAi Lines | Source | Stock number | Insertion site | Vector: | Other info: |
|---|---|---|---|---|---|
| *hsGFP::e2F* | This study | BL44206 | MI07660 (R) | | 3R:21,665,518 |
| Inactive Domains | | | | | |
| *hsGFP::Ets98B* | This study | BL54520 | MI10295 (R) | | 3R:27,745,742 |
| *hsGFP::Rbp6* | This study | BL38172 | MI04827 (F) | | 3L:17,149,391 |
| *hsGFP::OR67D* | This study | BL36157 | MI02878 (F) | | 3L:10,303,272 |
| *hsGFP::Tsp68C* | This study | BL43622 | MI07246 (F) | | 3L:13,842,947 |
| *hsGFP::Dpr6* | This study | BL44307 | MI06367 (F) | | 3L:9,984,992 |
| PcG domains | | | | | |
| *hsGFP::hmx* | This study | BL36161 | MI02896 (R) | | 3R:17,560,365 |
| *hsGFP::hmx* | This study | BL52065 | MI09049 (R) | | 3R:17,569,093 |
| *hsGFP::antp* | This study | BL33187 | MI02272 (F) | | 3R:6,908,787 |
| *hsGFP::bxd* | This study | BL60822 | MI09088 (R) | | 3R:16,794,996 |
| *hsGFP::rn* | This study | BL43893 | MI06953 (R) | | 3R:7,311,922 |
| *hsGFP::bab2* | This study | BL44856 | MI04190 (R) | | 3L:1,147,576 |
| *hsGFP::pb_1* | This study | BL41080 | MI05830 (F) | | 3R:6,698,519 |
| *hsGFP::pb_2* | This study | BL42174 | MI06772 (R) | | 3R:6,701,373 |
| *hsGFP::Blimp1* | This study | BL41087 | MI06053 (F) | | 3L:5,632,860 |
| *hsGFP::Blimp1_en* | This study | BL40186 | MI02744 (F) | | 3L:5,618,112 |
| *hsGFP::chinmo* | This study | BL51118 | MI08885 (R) | | 2L:1,647,426 |
| *hsGFP::klu* | This study | BL44148 | MI05554 (F) | | 3L:11,005,303 |

## Primary Antibody Table

| Epitope | Species | Source/Reference | Concentration for IF |
|---|---|---|---|
| H3K27me3 | rabbit | Cell Signaling C36B11 (#139619) Lot 14 | 1:1000 |
| H3K27me2 | rabbit | Cell Signaling D18C8 (#9728) Lot 15 | 1:1000 |
| H3K27me1 | rabbit | Millipore (#07–448) Lot 3233123 | 1:1000 |
| Orb | mouse | Developmental Studies 6H4 (*Lantz et al., 1994*) | 1:50 |
| Chinmo | rabbit | Tzumin Lee (*Zhu et al., 2006*) | 1:500 |
| H3K27ac | rabbit | Active Motif (#39135) Lot 17513002 | 1:1000 |
| Hts (1B1) | mouse | Developmental Studies (*Zaccai and Lipshitz, 1996*) | 1:50 |
| Ddx4 | mouse | Abcam (#ab27591) Lot GR290112-3 | 1:400 |
| H3K4me3 | rabbit | Cell Signaling C42D8 (#9751) Lot 7 | 1:1000 |
| ph | rabbit | Judy Kassis via Donna Arndt-Jovin (*Buchenau et al., 1998*) | 1:500 |
| pho | rabbit | Judy Kassis (*Brown et al., 2003*) | NA |
| Scm | rabbit | Jürg Müller (*Grimm et al., 2009*) | 1:1000 |
| Pcl | rabbit | Kevin White (KW4-PCL-D2) (*Riddle et al., 2012*) | 1:1000 |
| H2AK119ub | rabbit | Cell Signaling D27C4 (#8240) Lot 6 | 1:1000 |
| GFP | mouse | Invitrogen 3E6 (#A11120) Lot 764809 | 1:1000 |
| GFP mouse Invitrogen 3E6 (#A11120) Lot 764809 | | | 1:1000 |

## Oligos used in this study

| Oligo name | Sequence (5'-3') |
| --- | --- |
| MiMIC5'in GCGGCGTAATGTGATTTACTATC | MiMIC3'in |
| ACTAATGTAACGGAAGCTTCCCA | hsGFP5'out |
| GCTTGGTTATGCTTATCGTACCGA | hsGFP3'out |
| GAATTCGGTACCGGCGCGCCGAT | |
| wRNAiTop | CTAGCAGTGGAGCTTTCGCTCAGCAAATGTAGTT ATATTCAAGCATACATTTGCTGAGCGAAAGCTCCGCG |
| wRNAiBotom | AATTCGCGGAGCTTTCGCTCAGCAAATGTATGC TTGAATATAACTACATTTGCTGAGCGAAAGCTCCACTG |
| pclRNAiTop | CTAGCAGTACGATTCGAAACACACTTAAATAGTT ATATTCAAGCATATTTAAGTGTGTTTCGAATCGTGCG |
| pclRNAiBottom | AATTCGCACGATTCGAAACACACTTAAATATGCT TGAATATAACTATTTAAGTGTGTTTCGAATCGTACTG |
| pcRNAiTop | CTAGCAGTCGACGATCCAGTCGATCTAGTTAGT TATATTCAAGCATAACTAGATCGACTGGATCGTCGGCG |
| pcRNAiBottom | AATTCGCCGACGATCCAGTCGATCTAGTTATGC TTGAATATAACTAACTAGATCGACTGGATCGTCGACTG |
| sceRNAiTop | CTAGCAGTGCCTGGACATGCTGAAGAAGATAG TTATATTCAAGCATATCTTCTTCAGCATGTCCAGGCGCG |
| sceRNAiBottom | AATTCGCGCCTGGACATGCTGAAGAAGATATG CTTGAATATAACTATCTTCTTCAGCATGTCCAGGCACTG |
| TomF | CATGGTACCAACTTAAAAAAAAAAATCAAAAT GACTAGTAAGGGCGAGGAGGT |
| TomR | ATGTCTAGATTACTTGTACAGCTCGTCCATGC |

| Accession ChIP epitope or Input | | sample | | citation |
| --- | --- | --- | --- | --- |
| SRR1505727 | H3K27me3 | *Drosophila* embryo cycle 12/13 | | *Li et al., 2014* |
| SRR1505738 | Input | *Drosophila* embryo cycle 12/13 | | *Li et al., 2014* |
| SRR1505734 | Total H3 | *Drosophila* embryo cycle 12/13 | | *Li et al., 2014* |
| SRR1505728 | H3K27me3 | *Drosophila* embryo cycle 14 early | | *Li et al., 2014* |
| SRR1505739 | Input | *Drosophila* embryo cycle 14 early | | *Li et al., 2014* |
| SRR1505729 | H3K27me3 | *Drosophila* embryo cycle 14 late | | *Li et al., 2014* |
| SRR1505740 | Input | *Drosophila* embryo cycle 14 late | | *Li et al., 2014* |

### Reanalyzed public sequencing data

Transgenic fly construction

To construct pUASz-tdTomato-attB, we amplified tdTomato from pDEST-HemmarR (*Han et al., 2011*) with TomF + TomR and cloned it between the Acc65I and XbaI sites in pUASz-GFP-attB. To construct UASz-RNAi lines, we annealed Top and Bottom oligos and ligated the product between the NheI and EcoRI sites of pUASz-MiR. We introduced pUASz-tdTomato-attB into VK33 (3L:6442676) and all other attB-containing transgenes into attP40 (2L:5108448) using Rainbow Transgenics or BestGene Inc.

### Construction of hsGFP donor and introduction into existing MiMIC sites

GeneScript synthesized the hsGFP donor vector and we attached the sequence as a supplemental text file. hsGFP contains 400 bp upstream and 40 bp downstream of the Hsp70A transcription start site fused to a myosin intron, a synthetic translation enhancer, green fluorescent protein, and a P10 transcriptional terminator. We verified that both low- and high-expressing hsGFP inserts were insensitive to *hsp70*-derived piRNAs by test crosses to *hsp70Δ* as in *DeLuca and Spradling, 2018*. The

hsGFP reporter is flanked on both ends by tandem attB (for phiC31-mediated recombination with previously isolated MiMIC transposon insertions [*Venken et al., 2011*]) and FRT (for mobilizing the donor construct from its initial locus) sites. To create initial donor lines on the second and third chromosome, we introduced hsGFP randomly into the genome through its truncated P-element terminal repeats (100 bp 5' and 195 bp 3') and P-element transposase-encoded helper plasmid. We isolated positive transformants by identifying GFP fluorescing flies after heat shock. We selected 300 fly lines harboring MiMIC transposons, mostly on the 3rd chromosome to replace in parallel. We selected every available MiMIC insertion within PcG and Hp1-regulated domains, as well insertions equally spaced along length of the 3rd chromosome that sampled both active and inactive domains. To replace a MiMIC with our hsGFP reporter, we crossed a stable line carrying the hsGFP donor to flies carrying hsFLP, vasa-phiC31 integrase and appropriate marked chromosomes to generate F1 females carrying hsFLP, vasa-phiC31 integrase, the hsGFP donor, and appropriate marked chromosomes. We then crossed these F1 females to males carrying a yellow-marked MiMIC line of interest, and we heat-shocked the F2 eggs, larvae, and pupae at 37°C for 30 min every 2 days until adults eclosed. We then crossed F2 males carrying hsFLP, vasa-phiC31 integrase, the hsGFP donor, the yellow-marked MiMIC recipient and an opposing marked chromosome, to yellow mutant females. We screened F3 progeny for flies that carry the original MiMIC chromosome but are yellow in color (i.e. MiMIC insertions where *yellow* was replaced by *hsGFP*) and generated stable lines. Generally, we obtained at least one independent replacement for every five F2 males undergoing MiMIC replacement. The replacement success rate did not vary between MiMICs localized in different types of repressive chromatin, suggesting that germ cell precursors lack chromatin barriers to transgene insertion. We verified correct MiMIC replacement and determined the orientation of the reporter by PCR with primers flanking the hybrid attB/attP sites created by phiC31 recombinase. We attempted to replace 300 lines in parallel, using 20–30 F2 males for each line and successfully isolated at least one independent recombinant for 109 lines. We qualitatively examined hsGFP induction in all 109 lines, and because manual quantification was limited by sample processing and image acquisition time, we quantitated hsGFP induction for the 20 representative lines reported here.

## FACS sorting of live GSCs

We performed FACS sorting of live germline stem cells according to *Lim et al., 2012*. Briefly, we dissected *Vasa-GFP; bam*$^{\Delta86}$ ovaries in Grace's Media + 10% fetal bovine serum and rinsed 2x in phosphate buffered saline (PBS). We dissociated cells by incubating with 0.5% Trypsin and 2.5 mg/ml collagenase for 13 min at room temperature with intermittent vigorous shaking.

We washed 2x with PBS and twice filtered out large debris through a 50 µm nylon mesh filter. We resuspended cells in PBS + 1% BSA + 1 mM EDTA + 2 ng/µl propidium iodide and sorted GFP positive, PI negative cells on a BD FACSAria III running FACSDiva software. After sorting, we spun down cells and prepared total RNA using the Ambion mirVana miRNA isolation kit (#AM1560) according to the manufacturer's specifications without the miRNA enrichment step.

## FACS sorting nuclei for ChIP

We adapted a protocol from *Lilly and Spradling, 1996* to sort fixed, rather than live nuclei. We crossed *MTD-Gal4, UAS$_z$tdTomato* flies to *UAS$_z$-w$^{RNAi}$* (control) or *UAS-PcG$^{RNAi}$* flies to generate F1 progeny heterozygous for the *MTD-Gal4* drivers, *UAS$_z$tdTomato*, and *UAS-RNAi*. To collect GSC-like progenitor nuclei, we crossed *nanos-gal4, UAS$_p$-NLS-GFP, bam*$^{\Delta32}$ heterozygous flies to control *bam*$^{\Delta32}$ heterozygous flies or *UAS$_z$-Pcl$^{RNAi}$, bam*$^{\Delta32}$ heterozygous flies to generate F1 females homozygous for *bam*$^{\Delta32}$ and heterozygous for *nanos-gal4, UAS$_p$-NLS-GFP,* and for experimental samples, *UAS$_z$-Pcl$^{RNAi}$*. We fed 3–7 day-old adult F1 females wet yeast paste for 3 days in the presence of males and dissected batches of 30–60 ovaries in ice-cold PBS. We treated each batch with 5 mg/ml collagenase for 10 min at room temperature, and pipetted ovaries up and down to break up follicles and germaria. We rinsed 1x in PBS before fixing for 10 min in PBS + 2% paraformaldehyde at room temperature. We quenched fixation by adding 125 mM glycine for 5 min at room temperature, then washed in PBS. For *bam* ovary samples, we removed excess PBS, froze in liquid nitrogen, and stored at −80°C until we accumulated enough batches for our experiment. Before similarly freezing whole ovary samples, we passed the samples through a nylon mesh basket strainer to remove stage 10–14 follicles, which we found to interfere with subsequent fractionation steps. Once we accumulated

enough batches, we thawed and resuspended samples in nuclear isolation buffer (NIB) + 0.1 µg/ml DAPI: 15 mM TrisHCl 7.4, 60 mM KCl, 15 mM NaCl, 250 mM sucrose, 1 mM EDTA, 0.1 mM EGTA, 0.15 mM spermine, 0.5 mM spermidine, 1.5% NP40, protease inhibitor cocktail (Roche). We disrupted cells with 30 strokes in a Dounce homogenizer with the B pestle, placed the extract on top of a 1M/2M sucrose step gradient, and spun at 20,000 x g for 20 min. After removing the supernatant, we resuspended the nuclear pellet in NIB with 20–40 more strokes of the Dounce homogenizer with the B pestle (more strokes to dissociate smaller nuclei). We monitored nuclei dissociation after intermittent Douncing by visualizing DAPI under a fluorescence microscope. We passed nuclei through a 100 µm filter, and diluted with two volumes of PBS before sorting nuclei in on a BD FACSAria III cell sorter with 100 µm nozzle.

## Chromatin immunoprecipitation

For each IP, we started with a constant input of 0.5 million 2c-equivalent nuclei (for example, 0.5 million 2 c cells or 2000 256-c cells). When comparing different genotypes of a given stage, we added a spike-in of 2,000–10,000 similarly fixed mouse 3T3 tissue culture cells or FACS-isolated *Drosophila pseudoobscura* 2–16c follicle cells. We resuspended nuclei in 100 µl of 50 mM TrisHCl pH 8.0, 10 mM EDTA, 1% SDS, proteinase inhibitor cocktail (Roche) and sonicated in a Bioruptor Pico instrument with 22 cycles of 30 s on, 30 s off to fragment chromatin into mostly single nucleosome sizes (later confirmed with bioanalyzer). We then added 900 µl of dilution buffer: 15 mM TrisHCl pH 8.0, 1 mM EDTA, 1% Triton X-100, 150 mM NaCl and saved 1% of this extract as input. For each IP, we preincubated 10 µl of antibody with 25 µl of a 1:1 mix of proteinA:proteinG dynabeads and washed 2x with PBS + 0.02% Tween 20 (PBST). We combined antibody-conjugated beads with chromatin extracts on a rocker at 4°C overnight. Washed 2 × 15 min each with Wash buffer A: 20 mM TrisHCl pH8.0, 2 mM EDTA, 0.1% SDS, 1% Triton X100, 150 mM NaCl, Wash buffer B: 20 mM TrisHCl pH8.0, 2 mM EDTA, 0.1% SDS, 1% Triton X100, 500 mM NaCl, Wash buffer C:10 mM TrisHCl pH8.0, 1 mM EDTA, 1% NP40, 1% Sodium deoxycholate, 0.25M LiCl, TE buffer: 10 mM TrisHCl pH8.0, 1 mM EDTA. We eluted chromatin and reversed crosslinking by incubating at 65°C overnight with Direct Elution Buffer (DEB): 10 mM TrisHCl pH 8.0, 300 mM NaCl, 5 mM EDTA, 0.5% SDS. We additionally reversed crosslinks in input samples by adding NaCl to 300 mM and adding DEB to equalize the volume between inputs and IPs and incubating at 65°C overnight. We then treated samples for 30 min at 37°C with 0.3 mg/ml RNAse A, and 2 hr at 55°C with 0.6 mg/ml proteinase K, before extracting DNA with phenol:chloroform and precipitating with NaAc/ethanol. After a 70% ethanol wash, we resuspended samples in 10 µl water and used all 10 µl for library prep.

## RNA preparation for RNAseq

We crossed MTD-Gal4 females to UAS-RNAi males to generate F1 females heterozygous for the MTD-Gal4 drivers and a UAS-RNAi transgene. Because E(z)$^{GLKD}$ ovaries begin to degenerate at stage 6, we dissected ovaries from control Luciferase$^{GLKD}$ or E(z)$^{GLKD}$ females 0–8 hr after they eclosed from the pupal case. At this time point, both *control* and E(z)$^{GLKD}$ follicles had not progressed past stage 6 and therefore contained a nearly identical distribution of stages and cell types. The other PcG$^{GLKD}$ ovaries analyzed in this paper did not degenerate at a particular stage, so we compared RNA from fully developed ovaries in control w$^{GLKD}$ and PcG$^{GLKD}$. We fed 3–7 day old females wet yeast paste for 3 days before dissecting ovaries in cold PBS. We dissected 30 ovaries per replicate for ovaries containing nurse cell stages and 50 ovaries per replicate for *bam* mutant ovaries.

We purified total RNA using the TriPure reagent (Sigma Aldrich) according to the manufacturer's protocol.

## Library preparation and sequencing

For RNAseq, we used the Illumina TRUseq version 2 kit to create polyA-enriched mRNA libraries according to the manufacturer's specifications. For ChIPseq, we used the Takara Bio ThurPLEX DNA seq kit according to the manufacturer's specifications. Briefly, double stranded DNA ends are repaired, universal adapters are ligated on, and indexing is performed in a single tube using a total number of 15 PCR cycles. We sequenced both ChIP and RNAseq libraries on an Illumina NextSeq 500 using 75 bp single end reads.

## Ovary fixation for antibody staining and in situ hybridization

Before ovary fixation, we fed 3–7 day-old adult females wet yeast paste for 3 days in the presence of males. We dissected ovaries in PBS and fixed in PBS + 4% paraformaldehyde + 0.01% TritonX-100 for 13 min before washing 2x with PBS + 0.1% TritonX-100 (PTX).

## Antibody staining

We blocked fixed ovaries with PTX + 5% Normal Goat Serum (NGS) for 30 min. We incubated ovaries with primary antibody at 4°C overnight in PTX + 5% NGS, washed 3x in PTX, then incubated with alexa-fluor conjugated secondary antibody (1:1000) in NGS at 4°C overnight. We washed 3x in PTX, including 0.5 µg/ml DAPI in the second wash, and mounted in 50% glycerol. A list of primary antibodies and concentrations is listed in *Table 1*.

## In situ hybridization

We ordered custom Stellaris RNA FISH oligonucleotide probes directed against Pcl mRNA conjugated to CALFluor RED 590 from LGC Biosearch Technologies and performed in situ hybridization based on the company's recommendations. Briefly, we washed fixed ovaries 2 × 20 min. in Wash Buffer A (LGC Biosearch Technologies #SMF-WA1-60) + 10% formamide (WAF) then incubated ovaries for at least 2 hr in hybridization buffer (LGC Biosearch Technologies #SMF-HB1-10) + 10% formamide (HBF) at 37°C. We then incubated ovaries in HBF + 50 nM probe overnight, before washing 1x with HBF, 3x WAF at 37°, and 2x Wash Buffer B (LGC Biosearch Technologies #SMF-WB1-20) at room temperature. We mounted ovaries in 50% glycerol and imaged on a Leica SP8 scanning confocal.

## hsGFP reporter assay

We recombined each *hsGFP* reporter insertion with *NosGal4* using standard fly genetics. For each experiment, we crossed *hsGFP+NosGal4* females to *UAS-RNAi* males and assayed reporter induction in F1 progeny carrying the *hsGFP*, *NosGal4*, and *UAS-RNAi*. We cultured flies at 22°C to prevent premature activation of the heat-shock response and allow $E(z)^{GLKD}$ ovaries to produce rare follicles progressing past stage 6. For each experiment, we collected 3–7 day-old females and fed them wet yeast paste in the presence of males for 3 days to achieve a maximum rate of egg production. For each line, we heat shocked half of the well-fed flies while using the other half as a no heat shock control. We heat shocked flies in vials containing 1 mL of solidified 1% agar in a 37°C water bath for 20 min. After heat shock, we returned flies to normal food plus wet yeast for 3 hr to allow for GFP protein production and maturation. We dissected whole ovaries in PBS and fixed for 30 min in PBS + 4% paraformaldehyde + 0.01% Triton X-100. After one wash in PBS+0.1% TritonX-100 (PTX), we treated ovaries in PTX + 100 µg/ml RNAse A for 1–2 hr at room temperature before staining with 0.2 µg/ml Propidium Iodide to visualize DNA for developmental staging. We mounted ovaries in 50% glycerol and directly imaged GFP and Propidium Iodide fluorescence on a spinning disc confocal on a spinning disc confocal with 20x PlanApo 0.8 NA objective. We acquired a z-stack of confocal images and chose a single confocal plane through the middle of the desired germline stem cell or cyst and measured mean GFP fluorescence in germ cells in a manually drawn region of interest in image J. For each line and condition (+/- heat shock, different RNAi), we imaged at least three independent ovary pairs under identical laser power and acquisition settings and measured mean GFP fluorescence intensity in 1–5 germline stem cells or germ cell cysts of each stage per ovary pair. For each reporter line, to calculate a single experimental replicate for 'induction at a given stage', we first determined the mean intensity of all replicates of the non-heat shocked measurements for that stage and subtracted this mean from each experimental replicate for that stage.

## Genome segmentation

We converted a bed file of the nine state chromatin model for S2 cells from *Kharchenko et al., 2011* to *Drosophila* genome release 6.02 coordinates and simplified the nine chromatin state model into a four state model containing an active compartment (States 1–5) a PcG compartment enriched for PcG proteins and H3K27me3 (State 6), an inactive compartment enriched for H3K27me2 (State 9), and a Hp1 compartment enriched for Hp1a, Su(var)3–9, and H3K27me2/3 (State 7–8).

To annotate genes and transcription start sites, we used BEDTools (*Quinlan and Hall, 2010*) to intersect the four state model with the *Drosophila* genome release six annotation from Ensembl 81 to append a chromatin domain type to each protein-coding gene or transcription start site (TSS), removing any gene or TSS residing on the Y-chromosome, the 4th chromosome, or pericentric heterochromatin on any arm. Because genes have multiple isoforms that may span multiple domain types, we simplified the coordinates of the many isoforms of each gene into a single coordinate spanning the largest region shared by all isoforms. We then intersected these coordinates with the coordinates of PcG domains, retaining any gene with any bit of PcG domain as a PcG gene. We intersected the remaining genes with the coordinates of active domains, retaining any gene with any bit of an active domain as an active gene. We classified the remaining genes as inactive genes.

For ChIP analysis, we segmented the genome into five kilobase bins that start every 500 bases. We removed bins from the Y chromosome, the 4th chromosome, or any scaffolds not part of the remaining five chromosome arms. We additionally removed bins residing in the three most heavily amplified chorion gene clusters. We annotated the remaining bins with a single chromatin type by intersecting the list of bins with our four-state model, excluding any bin that contained multiple chromatin types from further analysis. Thus, the edges of chromatin domains were slightly underrepresented in our bin analyses. In *Figure 6*, we further classified PcG bins to separate bins that also contained a PRE. We classified PRE bins as any PcG bin intersecting with a composite list of pho plus ph peak summits called by MACS2.

## RNAseq analysis

We aligned 75 base pair single-end reads from at least 3 control and three experimental replicates to the *Drosophila* release six genome, Ensemble 81 annotation using the default parameters of Hisat2 (*Kim et al., 2015*). We measured raw transcripts per million for each gene using the default parameters of StringTie (*Pertea et al., 2015*). For differential expression analysis, we extracted raw read counts mapping to each gene, generated a DEseq2 model (*Love et al., 2014*), and analyzed and plotted normalized abundances of protein coding genes in R.

## ChIPseq analysis

We used bowtie2 (*Langmead and Salzberg, 2012*) to align 75 base pair single-end reads to either the *Drosophila melanogaster* release six genome (when no spike in was present) or a hybrid genome of *Drosophila melanogaster* and *Drosophila pseudoobscura* or *Mus musculus* (when corresponding spike in was present). We used the proportion of spike-in reads in Input and IP samples to generate a normalization factor for subsequent ChIPseq analysis and used a MAPQ 30 filter to remove ambiguously mapped reads. After scaling read coverage to reads per million and applying the normalization factor, we visualized and presented read depth across a genomic region of interest in the Integrative Genomics Viewer (*Robinson et al., 2011*), or generated genome wide summaries as described below.

To calculate read density across different chromatin domains, we used bedtools to assign spike-in normalized read coverage in Input and IP samples to annotated overlapping 5 kb bins. We used ggplot2 to plot a smoothed histogram of enrichments (IP/In) across all bins, with the y-axis corresponding to the proportion (not raw number) of bins for a given domain type. We used MACS2 (*Zhang et al., 2008*) to call peaks in Pho and Ph IP samples and Deeptools (*Ramírez et al., 2014*) to present raw read depth heat maps of genomic regions surrounding all peaks.

## Data and software availability

High throughput sequencing data can be found under accession GSE145282 and code is available at https://github.com/ciwemb/polycomb-development (*DeLuca, 2020*; copy archived at https://github.com/elifesciences-publications/polycomb-development).

## Acknowledgements

We thank Judy Kassis for Pho and Ph antibodies, Jürg Müller for Scm antibody, and Kevin White for Pcl antibody. We thank Patrick O'Farrell for Su(z)12-sfGFP, Zhao Zhang for UASp-NLS-GFP, and the Bloomington Stock Center and the *Drosophila* Gene Disruption project for other fly stocks. We thank Allison Pinder, Frederick Tan, and John Urban for help in generating and analyzing high

throughput sequencing data. SZD was supported by a Helen Hay Whitney Fellowship and MG was supported by a Jane Coffin Childs Fellowship.

## Additional information

### Funding

| Funder | Grant reference number | Author |
|---|---|---|
| Howard Hughes Medical Institute | ACS received the funding | Allan C Spradling |
| Helen Hay Whitney Foundation | Postdoc Fellowhip to Steven DeLuca | Steven Z Deluca |
| Jane Coffin Childs Memorial Fund for Medical Research | Postdoc Fellowship to Megha Ghildiyal | Megha Ghildiyal |

The funders had no role in study design, data collection and interpretation, or the decision to submit the work for publication.

### Author contributions

Steven Z DeLuca, Conceptualization, Resources, Data curation, Software, Formal analysis, Validation, Investigation, Visualization, Methodology, Writing - original draft; Megha Ghildiyal, Resources, Data curation, Investigation; Liang-Yu Pang, Resources, Data curation, Investigation, Visualization, Methodology; Allan C Spradling, Conceptualization, Supervision, Writing - review and editing

### Author ORCIDs

Steven Z DeLuca (iD) https://orcid.org/0000-0003-0683-8413
Liang-Yu Pang (iD) http://orcid.org/0000-0002-8373-5656
Allan C Spradling (iD) https://orcid.org/0000-0002-5251-1801

### Decision letter and Author response

Decision letter https://doi.org/10.7554/eLife.56922.sa1
Author response https://doi.org/10.7554/eLife.56922.sa2

## Additional files

### Supplementary files

• Transparent reporting form

### Data availability

High throughput sequencing data can be found under accession GSE145282 and code is available under an MIT license at https://github.com/ciwemb/polycomb-development (copy archived at https://github.com/elifesciences-publications/polycomb-development). Source data of image quantification is included as supporting files.

The following dataset was generated:

| Author(s) | Year | Dataset title | Dataset URL | Database and Identifier |
|---|---|---|---|---|
| Deluca SZ, Ghildiyal M, Liang-Yu P, Spradling AC | 2020 | Differentiating Drosophila female germ cells initiate Polycomb silencing by regulating PRC2-interacting proteins | http://www.ncbi.nlm.nih.gov/geo/query/acc.cgi?acc=GSE145282 | NCBI Gene Expression Omnibus, GSE145282 |

The following previously published dataset was used:

| Author(s) | Year | Dataset title | Dataset URL | Database and Identifier |
|---|---|---|---|---|
| Li XY, Harrison MM, Kaplan T, Eisen MB | 2014 | Establishment of regions of genomic activity during the Drosophila maternal-to-zygotic transition | https://www.ncbi.nlm.nih.gov/geo/query/acc.cgi?acc=GSE58935 | NCBI Gene Expression Omnibus, GSE58935 |

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
