## [Decision Letter]

**Acceptance summary:**

This paper describes an elegant new reporter system to study how Polycomb repression is established and maintained as germline stem cells differentiate into somatic cells. Leaning on the vast knowledge and diverse reagents uniquely available for manipulating the *Drosophila* ovary, the authors exploit this new system to identify factors that promote the initiation of Polycomb silencing.

**Decision letter after peer review:**

Thank you for submitting your article "Differentiating *Drosophila* female germ cells initiate Polycomb silencing by altering PRC2 sampling" for consideration by *eLife*. Your article has been reviewed by three peer reviewers, and the evaluation has been overseen by a Reviewing Editor and Jessica Tyler as the Senior Editor.

The reviewers have discussed the reviews with one another and the Reviewing Editor has drafted this decision to help you prepare a revised submission.

The editors have judged that your manuscript is of interest, but as described below extensive revising are is required before it is published. Note that we are currently offering, if you choose, to post the manuscript to bioRxiv (if it is not already there) along with this decision letter and a formal designation that the manuscript is 'in revision at *eLife*'. Please let us know if you would like to pursue this option.

Summary:

*Drosophila melanogaster* remains an important system for genetic, mechanistic, and biochemical investigations of Polycomb-mediated silencing and the consequences of Polycomb silencing on differentiation. De Luca et al. have established a new reporter system to study Polycomb-mediated silencing during *Drosophila* female germline development. The authors discover that PRC2-dependent H3K27me3 re-distributes from a broad presence in all inactive genes in germline stem cells to a sharp focus on PcG-silenced regions during differentiation into nurse cells. This redistribution mirrors H3K27 methylation observed in both fly early embryos and mouse ESCs. Importantly, the increase in PcG silencing coincides with an increase in Scm protein levels and a decrease in Pcl. Germline knockdown of these components suggest that increasing Scm protein level may cause localization of PcG proteins over PREs and increase PcG silencing. These data reveal new determinants of PcG silencing in the *Drosophila* ovary and possibly other systems.

The exploitation of transgenes embedded in alternative chromatin domains, studied in the context of the *Drosophila* ovary, offer new leverage to dissect Polycomb-mediated silencing. Moreover, the abundance of data gleaned from this new system represents an important contribution to the Polycomb field. However, there was consensus among the three reviewers that the current manuscript goes well-beyond the data in hand. For consideration in *eLife*, the manuscript should undergo a radical revision. The reviewers suggested that many inferences stated in the Results section and much of the Discussion, including but not limited to the sampling rate model and comparison to mammalian ES cells, and would be better suited to an Opinion piece. Moreover, the revision process should include hard decisions about which data offer a compelling interpretation and which data are better left for follow up in a future primary research manuscript. A substantially shorter manuscript, focused on the inducibility/repressibility of the reporter genes in different chromatin environments, could make the article suitable for publication in *eLife*. Furthermore, the title requires revision as well, term "sampling rate model" should be removed.

The reviewers noted that additional data, such as genome-wide H3K27ac profiles after RNAi of PRC2 components as well as knockdown verification using Western Blots, would strengthen the manuscript; however, given the current conditions, these experiments are not required for publication.

Essential revisions:

1) Radical re-writing that narrows the focus of the Results, that removes speculative models (including the sample rate model), and omits comparison to mammalian ES cells. See below for concerns raised by the current Discussion section.

2) The authors assert that *Drosophila* oogenesis offers an advantageous alternative to embryonic development to study PcG mechanisms. Moreover, comparison across *Drosophila* tissues and mouse ESCs suggest that indeed the pre-PcG silenced state in GSCs may share parallels with other systems. While the manuscript makes clear the advantages of this system for dissecting PcG silencing mechanisms, the reviewers were unconvinced that the new discoveries reported would necessarily extrapolate to other systems and instead may reflect special features of *Drosophila* oogenesis. Indeed, such germline-specific mechanisms have been shown for spermatogenesis. The authors should offer more balanced treatment of this issue.

3) The use of chromatin annotation from Karchenko et al. – which was not based on ovaries – weakens the designation of PcG domains for this system. It is well known that most sites that contain PRC1, PRC2, and H3K27me3 are not PcG-repressed until gastrulation. Are all of these sites PcG-silenced in the nurse cells? The genes involved will respond differently if they are being PcG repressed, if they do not bind PcG proteins at this stage, or if there are no activators at this stage. A more compelling rationale for parsing the genome based on these data is required.

4) Please remove or at least soften the following inferences in the Introduction and Results section:

a) Introduction: While PhoRC certainly contributes to recruitment, calling it the recruiter goes beyond the data. And in fact, as this work shows (Results), Pho binds to PREs in GSCs but does not recruit PRC1.

b) "PcG domains" contain PREs that recruit additional factors and bind PRC2 in a stable fashion, but nothing says that the repressive mechanism is different. Activating factors may very well be present around PREs, which have enhancers and promoters. In contrast, inactive regions may completely lack enhancers and promoters, making the comparison strained.

c) Inferences from the CBP section need revision or possibly removal. The reviewers did not agree that an increasing PRC2 antagonist explains the data.

5) Please offer explanations for the following in the Results section (or remove):

a) Please state in the main text how the efficiency of knockdown in the germline verified.

b) Why does repression decrease as nurse cells mature beyond stage 2? By stage 6 there are several PcG domains that are not repressed.

c) Figure 3B. The staining for H3K27me2, which should be the most abundant product of methylation, shows there is very little in the nurse cells (compare follicle cells). Please offer an explanation for this unexpected result.

For your revision of the Discussion, please consider the following concerns raised by reviewers (many of which may be moot after substantial revision/removal of Discussion content):

• The recruitment of PRC1 to PREs apparently begins as nurse cells begin to differentiate. The mechanism is not clear but it can hardly be by H3K27me3. Once PRE-focused recruitment of PRC1 and PRC2 begins, why not presume simply that PREs compete for PRC2 and there is less to go around?

• The Pcl^-^dependent residence time argument does not take into account other effects that target PRC2 such as the Esc-mediated binding to H3K27me3, interactions with PRC1, effects of nucleosome density, etc.

• Is there any reason why PRC2 freed of Pcl would silence inactive regions more effectively?

• The notion that H3K27me3 can recruit PRC1 by itself is poorly supported, especially in *Drosophila*, as the results in this paper also show, while the evidence shows that PRC1 can be recruited independently of PRC2.

• Bivalent marking of CpG islands in ESCs results from active recruitment of both H3K4 methylation and H3K27 methylation, not a sampling consequence.

• *Drosophila* has a higher density of PcG domains than mammals, not a higher number. This is more plausibly explained by the vast increase in noncoding sequences in mammals. Mammalian super enhancers are vastly larger and richer in regulatory elements than most *Drosophila* regulatory regions

• Like *Drosophila*, mammalian ESCs have a very high content of H3K27me2, arguing for considerable roaming.

---

## [Author Response]

Summary:*Drosophila melanogaster* remains an important system for genetic, mechanistic, and biochemical investigations of Polycomb-mediated silencing and the consequences of Polycomb silencing on differentiation. De Luca et al. have established a new reporter system to study Polycomb-mediated silencing during Drosophila female germline development. The authors discover that PRC2-dependent H3K27me3 re-distributes from a broad presence in all inactive genes in germline stem cells to a sharp focus on PcG-silenced regions during differentiation into nurse cells. This redistribution mirrors H3K27 methylation observed in both fly early embryos and mouse ESCs. Importantly, the increase in PcG silencing coincides with an increase in Scm protein levels and a decrease in Pcl. Germline knockdown of these components suggest that increasing Scm protein level may cause localization of PcG proteins over PREs and increase PcG silencing. These data reveal new determinants of PcG silencing in the Drosophila ovary and possibly other systems.The exploitation of transgenes embedded in alternative chromatin domains, studied in the context of the *Drosophila* ovary, offer new leverage to dissect Polycomb-mediated silencing. Moreover, the abundance of data gleaned from this new system represents an important contribution to the Polycomb field. However, there was consensus among the three reviewers that the current manuscript goes well-beyond the data in hand. For consideration in eLife, the manuscript should undergo a radical revision. The reviewers suggested that many inferences stated in the Results section and much of the Discussion, including but not limited to the sampling rate model and comparison to mammalian ES cells, and would be better suited to an Opinion piece. Moreover, the revision process should include hard decisions about which data offer a compelling interpretation and which data are better left for follow up in a future primary research manuscript. A substantially shorter manuscript, focused on the inducibility/repressibility of the reporter genes in different chromatin environments, could make the article suitable for publication in eLife. Furthermore, the title requires revision as well, term "sampling rate model" should be removed.

We “radically” reorganized, rewrote and shortened the Results and Discussion sections with the focus requested. We changed the title, and removed the name, “the sampling rate model,” but we still have the responsibility to interpret our data. However, our discussion of mechanism is now confined to the Discussion section where reference is made to a model figure.

The reviewers noted that additional data, such as genome-wide H3K27ac profiles after RNAi of PRC2 components as well as knockdown verification using Western Blots, would strengthen the manuscript; however, given the current conditions, these experiments are not required for publication.

We agree that genome wide H3K27ac profiles in PRC2 RNAi would complement and potentially be more sensitive than RNAseq, and could more directly address the model that H3K27methylation blocks acetylation. We cite Ferrari et al., 2014 and Lee et al., 2015 as examples of this experiment and support for the acetylation blocking model.

Essential revisions:1) Radical re-writing that narrows the focus of the Results, that removes speculative models (including the sample rate model), and omits comparison to mammalian ES cells. See below for concerns raised by the current Discussion section.

We removed any mention of ES cells from the Results or Discussion. We removed the name “sampling model” from the title, section headings, and text. However, we think that the data is consistent with the idea that reduced PRC2 sampling is why wild type GSCs lack silencing but PclGLKD GSCs have silencing and an altered histone methylation profile. We rewrote our argument to clarify our model and also noted that it is speculative. We think we should be allowed to publish a model that fits the data in the Discussion section, especially if an alternative model is not better

2) The authors assert that Drosophila oogenesis offers an advantageous alternative to embryonic development to study PcG mechanisms. Moreover, comparison across *Drosophila* tissues and mouse ESCs suggest that indeed the pre-PcG silenced state in GSCs may share parallels with other systems. While the manuscript makes clear the advantages of this system for dissecting PcG silencing mechanisms, the reviewers were unconvinced that the new discoveries reported would necessarily extrapolate to other systems and instead may reflect special features of *Drosophila* oogenesis. Indeed, such germline-specific mechanisms have been shown for spermatogenesis. The authors should offer more balanced treatment of this issue.

We removed direct speculation of how our results could apply to ESCs. We mostly discussed our results in the context of germ cell and early embryo chromatin as germ cells are the most fundamentally conserved lineage throughout all animals. We also discussed why spermatogenesis or mammalian oocytes may be different.

3) The use of chromatin annotation from Karchenko et al. – which was not based on ovaries – weakens the designation of PcG domains for this system. It is well known that most sites that contain PRC1, PRC2, and H3K27me3 are not PcG-repressed until gastrulation. Are all of these sites PcG-silenced in the nurse cells? The genes involved will respond differently if they are being PcG repressed, if they do not bind PcG proteins at this stage, or if there are no activators at this stage. A more compelling rationale for parsing the genome based on these data is required.

We produced 2 new figure panels (Figure 2E,F), Table 1, and a revised Results sections to show the extensive conservation between PcG domains in nurse cells and somatic cells. We stuck with Karchenko et al. because we needed a “standard” to compare cell types (embryo, follicle cell, nurse cell, S2cell) anyway and we want to emphasize that nurse cells are extremely similar to any somatic cell. We think that inventing a new nurse cell specific annotation will give readers the incorrect impression that nurse cells are highly divergent from somatic cells and that our system is not valuable for understanding somatic cell Polycomb silencing.

4) Please remove or at least soften the following inferences in the Introduction and Results section:a) Introduction: While PhoRC certainly contributes to recruitment, calling it the recruiter goes beyond the data. And in fact, as this work shows (Results), Pho binds to PREs in GSCs but does not recruit PRC1.

We removed reference to Pho being “the recruiter”.

b) "PcG domains" contain PREs that recruit additional factors and bind PRC2 in a stable fashion, but nothing says that the repressive mechanism is different. Activating factors may very well be present around PREs, which have enhancers and promoters. In contrast, inactive regions may completely lack enhancers and promoters, making the comparison strained.

We agree that the repressive mechanism may be the same and simplified our writing in serval places to address this point.

c) Inferences from the CBP section need revision or possibly removal. The reviewers did not agree that an increasing PRC2 antagonist explains the data.

We removed the entire CBP section as we agree it is not essential for the paper. We additionally included a new RNAseq analysis (Figure 7B) arguing that PRC2-dependent repression, and not activation, primarily increase the disparity between active and inactive gene expression during development

5) Please offer explanations for the following in the Results section (or remove):a) Please state in the main text how the efficiency of knockdown in thegermline verified.

We clarify how we verified knockdown and refer to the main figure panels.

b) Why does repression decrease as nurse cells mature beyond stage 2? By stage 6 there are several PcG domains that are not repressed.

This concern is not consistent with the data and may be due to confusion about how repression is detected. Repression is not a comparison of the overall reporter induction in one stage versus another stage. Repression is measured in two ways: the induction difference between a repressed reporter and an active reporter at the same stage (Figure 1G) or the induction difference between control and E(z)GLKD for a single reporter at a single stage (Figure 1H). By these measures, repression increases throughout oogenesis. We added several statements to make this clearer in the Results section.

c) Figure 3B. The staining for H3K27me2, which should be the most abundant product of methylation, shows there is very little in the nurse cells (compare follicle cells). Please offer an explanation for this unexpected result.

We don’t know why this is. We revised this Results section to bring this to the readers’ attention. For this paper, we think it is more important to compare germ cell nuclei from different stages anyway.

For your revision of the Discussion, please consider the following concerns raised by reviewers (many of which may be moot after substantial revision/removal of Discussion content):• The recruitment of PRC1 to PREs apparently begins as nurse cells begin to differentiate. The mechanism is not clear but it can hardly be by H3K27me3. Once PRE-focused recruitment of PRC1 and PRC2 begins, why not presume simply that PREs compete for PRC2 and there is less to go around?

We agree that PRC1 recruitment is likely not by H3K27me3. We’re not sure what the reviewers mean by “Once PRE-focused recruitment of PRC1 and PRC2 begins, why not presume simply that PREs compete for PRC2 and there is less to go around?” We agree that PREs deplete PRC2 from inactive regions. But inactive regions actually gain silencing, not lose it. We think this is extremely interesting and was part of the basis for our model. How could less PRC2 have more silencing capability in inactive domains? We propose it samples faster and methylates more nucleosomes. We tried to explain this in more simple terms in the completely rewritten Discussion section

• The Pcl^-^dependent residence time argument does not take into account other effects that target PRC2 such as the Esc-mediated binding to H3K27me3, interactions with PRC1, effects of nucleosome density, etc.

The Pcl residence time argument is based on in vitro data from Choi et al. and our PclGLKD phenotype in GSCs. We agree that other mechanisms affect PRC2 activity and H3K27me abundance and we now discuss the effect of Esc. Also, our model does not discount that these forms of regulation exist, it simply adds to them.

• Is there any reason why PRC2 freed of Pcl would silence inactive regions more effectively?

The surprising result that Pcl inhibits reporter silencing in GSCs is the foundation of the sampling model. Region 1 germ cells and nurse cells are constantly replicating their DNA and incorporating new histones. We propose that Pcl^-^PRC2 cannot sufficiently sample these new histones in inactive domains due to its long residence time on DNA. We propose incomplete sampling could result in many inactive nucleosomes being unmethylated. We rewrote the Discussion to lay out our arguments for using histone methylation to infer PRC2 sampling rates and we present our evidence that core-PRC2 does most of the inactive domain silencing by sampling faster than Pcl^-^PRC2.

• The notion that H3K27me3 can recruit PRC1 by itself is poorly supported, especially in *Drosophila*, as the results in this paper also show, while the evidence shows that PRC1 can be recruited independently of PRC2.

We didn’t mean that H3K27me3 exclusively recruits PRC1. We removed this confusing statement.

• Bivalent marking of CpG islands in ESCs results from active recruitment of both H3K4 methylation and H3K27 methylation, not a sampling consequence.

We removed this section but we are afraid the reviewers did not understand the point we were trying to make. We were speculating that under-sampling at CGIs could generate a mixed population of cells- some with H3K27 methylation and some with H3K4 methylation. The idea that bivalency is actually bistability is not ours, for example see Sneppen and Ringrose, 2019. We were just proposing that under-sampling can generate a mixed population of cells. We are unaware of a single complex that actively targets H3K27me3 and H3K4me3 to the same nucleosome and thus an “active mechanism for bivalency.” Of course there are active mechanisms that independently target both mods and these are compatible with the bistability and undersampling models.

• *Drosophila* has a higher density of PcG domains than mammals, not a higher number. This is more plausibly explained by the vast increase in noncoding sequences in mammals. Mammalian super enhancers are vastly larger and richer in regulatory elements than most Drosophila regulatory regions

We eliminated all of this discussion.

• Like *Drosophila*, mammalian ESCs have a very high content of H3K27me2, arguing for considerable roaming.

We removed these sections as our arguments are better laid out in an extended review format.